# C14ORF39/SIX6OS1 is a constituent of the synaptonemal complex and is essential for mouse fertility

Laura Gómez-H[1,*], Natalia Felipe-Medina[1,*], Manuel Sánchez-Martín[2,3], Owen R. Davies[4], Isabel Ramos[1], Ignacio García-Tuñón[1], Dirk G. de Rooij[5], Ihsan Dereli[6], Attila Tóth[6], José Luis Barbero[7], Ricardo Benavente[8], Elena Llano[1,9] & Alberto M. Pendas[1]

Meiotic recombination generates crossovers between homologous chromosomes that are essential for genome haploidization. The synaptonemal complex is a 'zipper'-like protein assembly that synapses homologue pairs together and provides the structural framework for processing recombination sites into crossovers. Humans show individual differences in the number of crossovers generated across the genome. Recently, an anonymous gene variant in *C14ORF39/SIX6OS1* was identified that influences the recombination rate in humans. Here we show that *C14ORF39/SIX6OS1* encodes a component of the central element of the synaptonemal complex. Yeast two-hybrid analysis reveals that SIX6OS1 interacts with the well-established protein synaptonemal complex central element 1 (SYCE1). Mice lacking SIX6OS1 are defective in chromosome synapsis at meiotic prophase I, which provokes an arrest at the pachytene-like stage and results in infertility. In accordance with its role as a modifier of the human recombination rate, SIX6OS1 is essential for the appropriate processing of intermediate recombination nodules before crossover formation.

[1] Instituto de Biología Molecular y Celular del Cáncer (CSIC-Universidad de Salamanca), 37007 Salamanca, Spain. [2] Departamento de Medicina, Universidad de Salamanca, 37007 Salamanca, Spain. [3] Transgenic Facility, Nucleus platform, Universidad de Salamanca, 37007 Salamanca, Spain. [4] Institute for Cell and Molecular Biosciences, Newcastle University, Newcastle upon Tyne NE2 4HH, UK. [5] Reproductive Biology Group, Division of Developmental Biology, Department of Biology, Faculty of Science, Utrecht University, 3584CM Utrecht, The Netherlands. [6] Institute of Physiological Chemistry, Medical Faculty of TU Dresden, Fiedlerstrasse 42, 01307 Dresden, Germany. [7] Departamento de Biología Celular y Molecular, Centro de Investigaciones Biológicas (CSIC), Madrid 28040, Spain. [8] Department of Cell and Developmental Biology, Biocenter, University of Würzburg, D-97074 Würzburg, Germany. [9] Departamento de Fisiología y Farmacología, Universidad de Salamanca, 37007 Salamanca, Spain. * These authors contributed equally to this work. Correspondence and requests for materials should be addressed to A.M.P. (email: amp@usal.es) or to E.L. (email: ellano@usal.es).

During meiosis, two successive rounds of chromosome segregation occur following a single round of replication, resulting in the formation of haploid gametes from diploid progenitors[1]. This ploidy reduction is achieved through a series of meiosis-specific events, including pairing, synapsis, crossover formation between homologues, suppression of sister centromere separation during the first (reductional) division and separation of sister chromatids during the second (equational) division. Homologous chromosomes become tethered together through numerous recombination events between homologous non-sister chromatids, which are triggered by double-strand break induction. Through resolution, a subset of recombination events mature into crossovers (chiasmata) that maintain the physical tethering between homologues until the onset of anaphase I (ref. 1).

In humans, the number of crossovers occurring across the genome differs between individuals. Through exploitation of data resources in Iceland, Kong et al.[2] recently analysed over two million recombination events and putative variants from 2,261 whole genome–sequenced individuals to identify variants that influence the global recombination rate. Among the new variants, several coding SNPs in very well-known meiotic players were identified, including the histone methyltransferase PRDM9 and the meiotic cohesin RAD21L, the latter of which has been the focus of our previous meiotic studies[3–5].

In addition to their well-established role in mediating sister chromatid cohesion through ring structure formation, cohesin complexes are also responsible for the assembly of the synaptonemal complex (SC)[5–7]. The SC is a proteinaceous structure that holds homologous chromosome pairs in synapsis during prophase I, from zygonema to pachynema. It consists of two parallel axial elements (AEs) that bind sister chromatids together, and which become known as lateral elements (LEs) upon chromosome pairing. It also contains transverse filaments, which connect (synapse) the two LEs together. Transverse filament proteins are recruited to LEs and undergo zipper-like assembly, bridging between LEs through the formation of the midline central element (CE), and thereby generating the tripartite structure of the SC[8]. To date, only seven protein structural components of the SC have been identified in mammals, namely LE proteins SYCP2 and SYCP3, transverse filament protein SYCP1, and CE proteins SYCE1, SYCE2, SYCE3 and TEX12 (ref. 9). The location of CE-specific proteins is, by definition, restricted to the synapsed regions of the chromosomes from zygotene to diplotene[9]. The SC provides the structural framework for synapsis, double-strand break (DSB) repair and exchange between homologues[10,11]. It is known from mouse mutants and through human genetic analysis of families with non-obstructive azoospermia and premature ovarian failure, that alterations in these genes (that is, meiosis-specific cohesin subunit STAG3, and SYCE1) can result in meiotic arrest and human infertility[12,13].

To gain further insight into the biological processes affecting recombination rates across the human genome, we have investigated the list of genes that were recently identified as having coding variants[2]. We focus our analysis on the anonymous C14ORF39/SIX6OS1 gene (herein SIX6OS1) based on its restricted pattern of transcription and expression. Here, we show that C14ORF39/SIX6OS1 encodes a component of the CE of the SC. Yeast two-hybrid analysis reveals that SIX6OS1 interacts with SYCE1. In addition, mice lacking SIX6OS1 are defective in chromosome synapsis at meiotic prophase I, which provokes an arrest at the pachytene-like stage and results in mouse infertility. In accordance with its role as a modifier of the human recombination rate, SIX6OS1 is essential for the appropriate processing of intermediate recombination nodules before crossover formation in mice.

## Results

**C14ORF39/SIX6OS1 is a protein of the mammalian SC.** The sequence variants identified by Kong et al.[2] include known genes functioning in meiotic recombination such as RNF212 (refs 14,15), RAD21L (ref. 4), PRDM9 (ref. 16), MSH4 (ref. 17) and CCNB1P1 (ref. 18). They also include an anonymous open reading frame, containing a nonsynonymous SNV with unknown function (rs1254319, p.Leu524Phe). This gene, named SIX6OS1, is also annotated as coding for a natural antisense transcript (NAT) that is associated with the eye transcription factor SIX6 (ref. 19). However, and in contrast to most natural antisense transcripts, SIX6OS1 shows a high degree of sequence similarity between mouse (4930447C04Rik) and human (C14ORF39), and contains large theoretical conserved open reading frames encoding putative proteins of 587 and 574 residues in mouse and human, respectively (Supplementary Fig. 1). Phylogenetic analysis indicates that SIX6OS1 is a unique gene that appeared firstly in the genomes of cartilaginous fish, and it can be clearly identified from lobed fin fish to mammals (Supplementary Fig. 1). Interestingly, in the variant rs1254319 (p.Leu524Phe), the phenylalanine residue is very well conserved in all genomes (including several other primates, for example, Orangutan and Baboon) except humans (Supplementary Fig. 1).

Analysis of Six6os1 mRNA expression in mouse tissues by RT–qPCR (Fig. 1a) revealed that it is most abundantly expressed in testis (in agreement with GTEx database[20]).

The Six6os1 open reading frame predicts a protein of around 70 kDa, in agreement with our western blot analysis (Supplementary Fig. 3a). Sequence analysis reveals the presence of an evolutionarily conserved region of high helical content within its N terminus (corresponding to amino acids 1–261), including a short stretch of predicted coiled-coil structure towards the C-terminal end of this region (Supplementary Fig. 2a). These features are typical of SC proteins, which commonly contain a high percentage of helical content and adopt homo- or hetero-oligomeric helical bundle or coiled-coil structures[21,22]. The presence of conserved proline residues between predicted helices suggests that the structure includes helix–turn–helix motifs, rather than adopting an extended helical conformation such as that observed in the crystal structure of SYCP3 (ref. 22). This feature is in common with SC central element proteins SYCE1 and SYCE3, but contrasts with the elongated helical structure predicted and observed in solution for central element complex SYCE2-TEX12 and transverse filament protein SYCP1 (ref. 21). We therefore predict that this N-terminal helical region could mediate interactions with structural proteins of the SC. The remainder of the SIX6OS1 sequence is predicted to be largely unstructured, but importantly contains patches of evolutionary conservation towards its C-terminal end (Supplementary Fig. 2b). These features are characteristic of flexible sequences that interact with globular proteins at specific peptide motifs through induced fit, and thereby mediate the assembly of macromolecular complexes. The unstructured C-terminal region further contains numerous predicted phosphorylation sites, including four conserved S/TP potential CDK phosphorylation sites (Supplementary Fig. 2c), which may function in regulating the timely assembly of such macromolecular complexes during the first meiotic division.

To explore the localization of SIX6OS1, we in vivo electroporated an expression plasmid encoding SIX6OS1-GFP[23] into mouse testis. After 48 h, SIX6OS1-GFP co-localized with SYCP3 along the synapsed LEs at pachynema (spermatocytes in which homologues are fully synapsed) (Fig. 1b). In addition, we carried out a detailed analysis of mouse spermatocytes and oocytes spreads through double labelling with specific antibodies against SIX6OS1 (which were intensively validated, Fig. 5c;

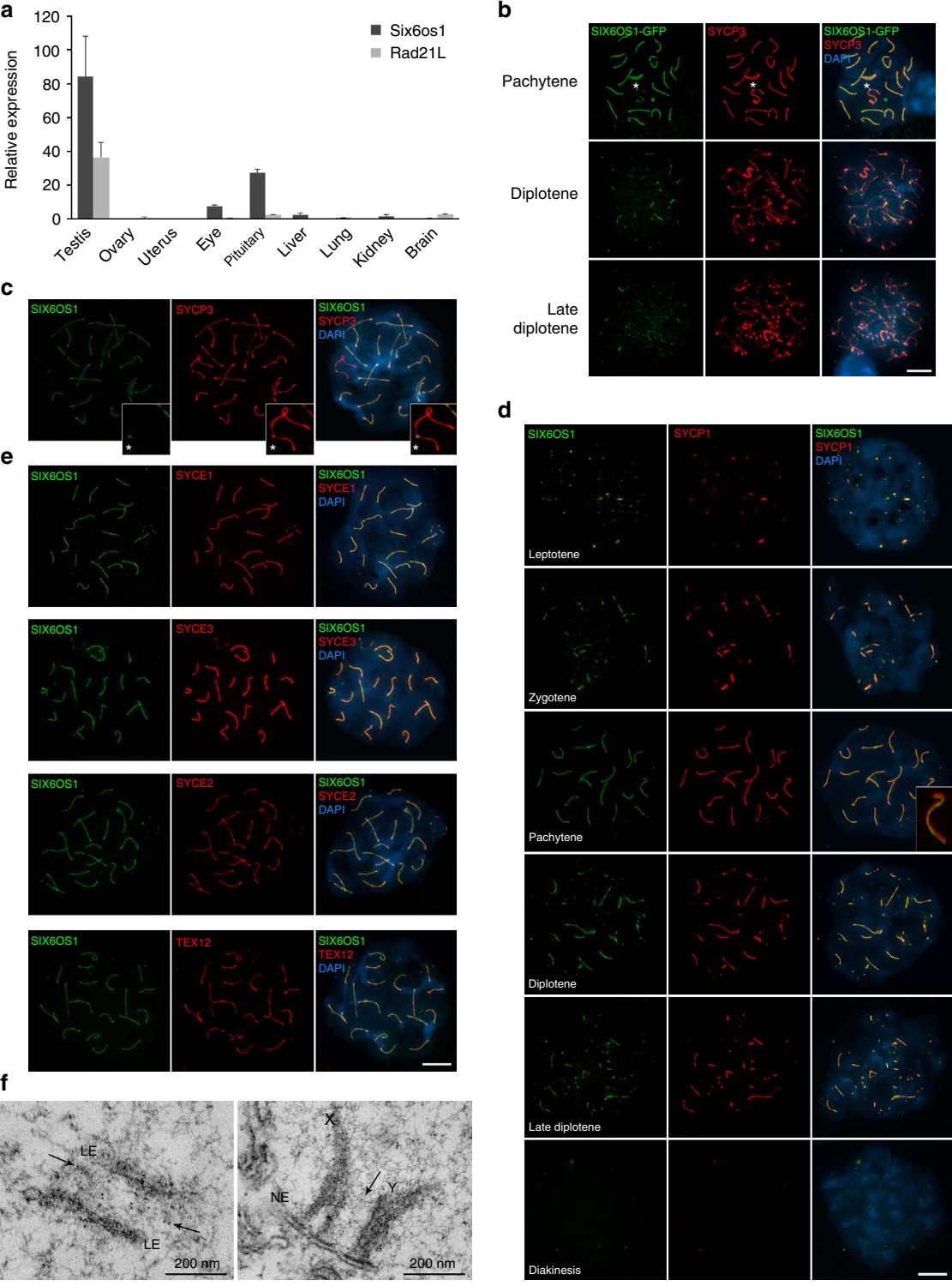

**Figure 1 | Transcriptional analysis and distribution of SIX6OS1 in mouse meiocytes. (a)** Relative transcription of *Six6os1* and *Rad21l* (ref. 4) mRNA by quantitative reverse transcription PCR (RT–qPCR) in mouse tissues. β-Actin transcription was used to normalize the expression (mean ± s.d., three replicates). (**b**) Immunolabelling of *in vivo* electroporated SIX6OS1-GFP in mouse testis. SIX6OS1 was detected with anti-GFP (green) and endogenous SYCP3 was detected using mouse anti-SYCP3 (red). DNA was stained with DAPI (blue). During pachytene, SIX6OS1 colocalizes with SYCP3 along synapsed lateral elements (LEs) including the pseudoautosomal region (PAR) of the XY bivalent (spermatocytes). In diplotene and late diplotene, SIX6OS1 localizes at the still synapsed LEs. (**c**) Double immunolabelling of endogenous SIX6OS1 (green) and SYCP3 (red) in spermatocytes. DNA was stained with DAPI (blue). During pachynema, SIX6OS1 is located at the synapsed autosomal LEs and at the PAR of the sex XY bivalent. (**d,e**) Co-labelling of spermatocytes spread preparations with SIX6OS1 (green) and SYCP1, SYCE1, SYCE3, SYCE2 or TEX12 (red), showing that SIX6OS1 localizes to the synapsed LEs but best mirrors SYCE1 localization. (**f**) Immunoelectron microscopy of frozen mouse testis sections marked with goat anti-SIX6OS1 antibody. Left panel corresponds to an autosomal chromosome and right panel to the XY bivalent in which the PAR is shown. Gold particles 6 nm. Scale bar in **b**–**e**, 10 μm. PAR is indicated with an asterisk in **b** and **c**.

Supplementary Fig. 3) and SYCP3 or SYCP1 (Fig. 1c,d; Supplementary Fig. 4). SIX6OS1 was detected from zygonema to pachynema, co-localizing with SYCP1 along synapsed LEs, but with diminished co-localization at telomeres (Fig. 1d). On the XY bivalent, the pseudoautosomal synapsed region labelled positively for SIX6OS1 (Fig. 1c; Supplementary Fig. 4b). As desynapsis

progressed through diplonema, SIX6OS1 (together with SYCP1) was not observed at the desynapsed regions of spermatocytes and oocytes (Fig. 1d; Supplementary Fig. 4a). Thus, SIX6OS1 partially overlaps the distribution of SYCP1 at the synapsed axes.

We next measured and compared the fluorescence profile of SIX6OS1 along the chromosome axes with those of CE

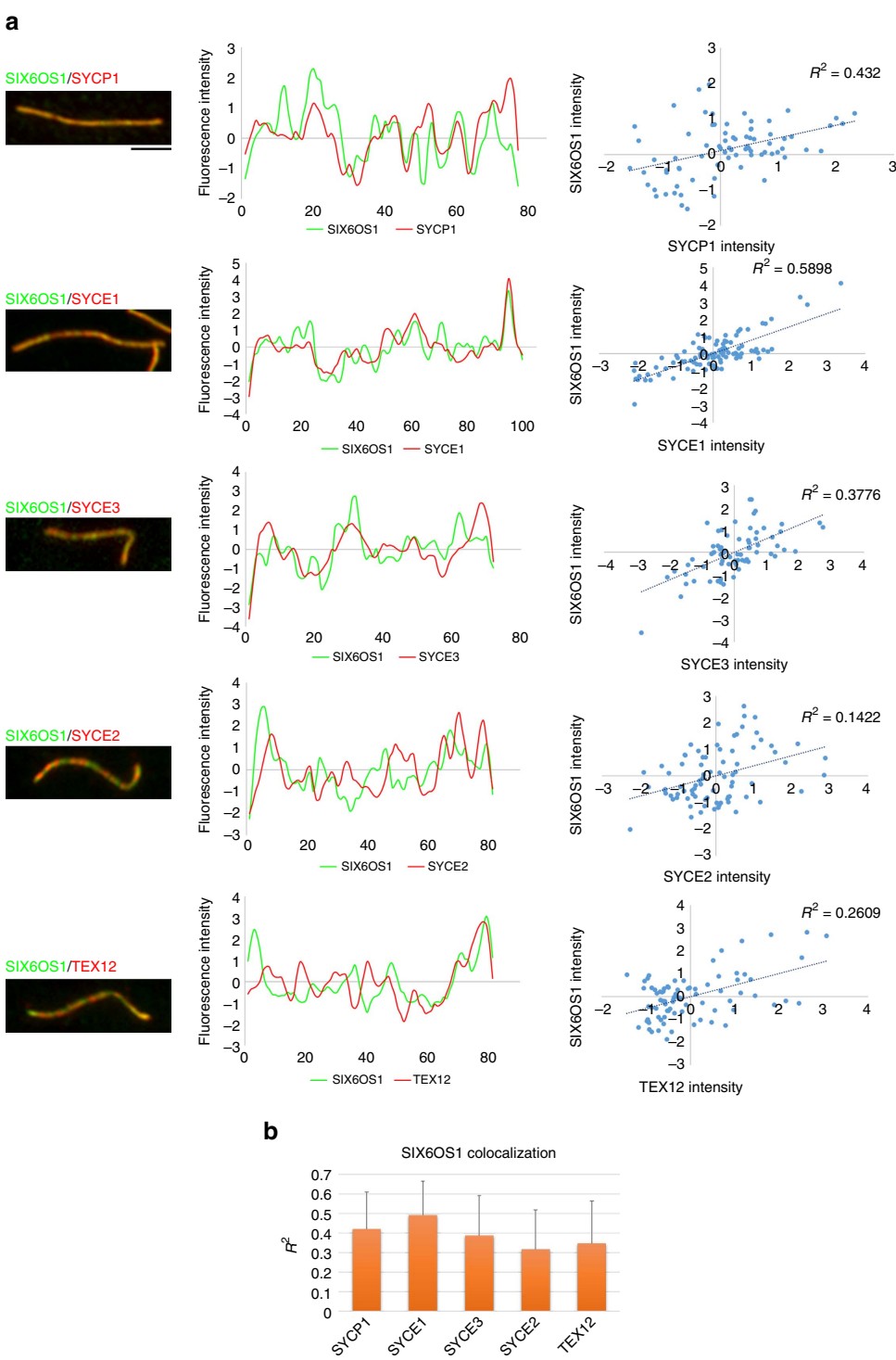

**Figure 2 | Co-localization profile of SIX6OS1 with central element proteins.** (**a**) Double immunostaining of SIX6OS1 (green) and SYCP1, SYCE1, SYCE3, SYCE2 or TEX12 (red). Immunofluorescence signal levels were measured on synapsed chromosome axes. Left plots represent normalized signal intensity profiles of SIX6OS1 with each CE protein. Right plots show regression analysis of the correlation between each pair. (**b**) Plot of the mean correlation between SIX6OS1 and SYCP1, SYCE1, SYCE3, SYCE2 or TEX12 ($n = 38$ axial elements (AEs), mean ± s.d.). The best correlation value was obtained with SYCE1. Scale bar, 2.5 μm.

proteins SYCP1, SYCE1, SYCE2, SYCE3 and TEX12 (Figs 1e, 2a). This revealed that SIX6OS1 localization is more similar to the continuous pattern of SYCP1 and SYCE1/3 (best regression with SYCE1) than to the more punctate pattern of SYCE2 and TEX12 (ref. 24) (Fig. 2b). However, the SIX6OS1 localization pattern is not strictly identical to SYCP1, especially at the telomeres, where SIX6OS1 stained more weakly (Figs 1d and 2a).

In addition, we performed immuno-gold electron microscopy on testis sections using the same SIX6OS1 antibody. The gold particle distribution agrees with those previously reported for CE proteins SYCE1, SYCE2, TEX12 and SYCE3 (refs 24,25) thus supporting the localization of SIX6OS1 at the CE (Fig. 1f). Taken together, these results demonstrate that SIX6OS1 is a meiotic protein that is located at the CE of the SC.

**SIX6OS1 interacts with SYCE1.** To understand the role of SIX6OS1 in meiosis, we searched for proteins that interact with mouse SIX6OS1 through yeast two hybrid (Y2H) screening (see Methods). Of the 6.1 million independent clones screened, 90 colonies containing interacting bait and prey fusion proteins grew under the highest stringency conditions. Analysis of the positively interacting clones (Methods) revealed that they all encode SYCE1, a well-known protein of the CE of the SC[26]. To confirm and validate this interaction, we made use of heterologous HEK 293T cells by transiently transfecting expression plasmids encoding GFP-SIX6OS1 and Myc-SYCE1. SIX6OS1 was found to co-immunoprecipitate (co-IP) reciprocally with SYCE1 (Fig. 3a).

Through biochemical, biophysical and crystallographic studies, all SC proteins studied to date have proven to exist as homo- and/or hetero-oligomers. To explore the possible self-association of SIX6OS1, we co-transfected Six6os1 tagged with two different epitopes (GFP and Flag) and found that they co-immunoprecipitate (Fig. 3b), suggesting that it exists as a homo-oligomer.

Next, we adopted a candidate gene approach to identify additional putative interactors of SIX6OS1. We co-transfected Six6os1 with cDNAs encoding each of the known central element proteins (SYCE1, SYCE2, SYCE3 and TEX12), transverse filament protein SYCP1, LE protein SYCP3, and the meiotic cohesins REC8 and Sororin (a recently identified cohesin subunit localized to synapsed regions[27]) (Fig. 3a; Supplementary Fig. 5a,b). As positive controls we used the well-known interaction between SYCE2 and TEX12 (ref. 21), and between SYCE3 and SYCE1 (ref. 28; Supplementary Fig. 5c). We detected co-immunoprecipitation only between SIX6OS1 and SYCE1.

Finally, we used truncated forms of SIX6OS1 to show that the N-terminal half (1–286), but not the C-terminal half (287–574), is able to interact with SYCE1 in isolation (Fig. 3c). Together, these results indicate that the interaction between SIX6OS1 and SYCE1 occurs in a very specific manner through the N-terminal half of SIX6OS1.

**Polycomplex formation of SIX6OS1.** SYCP1 and SYCP3 form filamentous structures, so-called polycomplexes, in the cytoplasm of transfected cells. Thus, co-expression of an interacting partner with SYCP1 or SYCP3 may lead to its recruitment to polycomplexes[25,29]. To analyse this, we transfected the cDNA encoding SIX6OS1 in combination with SYCP1 alone or in different combinations with SYCE1, SYCE2, SYCE3 and TEX12, and studied their distribution by immunofluorescence. In absence of SYCP1, single transfections of SYCE1, SYCE3, SYCE2, TEX12 and SIX6OS1 produced different distributions (cytoplasmic aggregates, whole cell, cytoplasmic and nuclear, respectively), but in all cases without the appearance of self-assembled higher order structures (Fig. 3d; Supplementary Fig. 6a).

When transfected in combination with SYCP1, SIX6OS1 was not recruited to the filamentous structures, and its cellular localization was not modified (Fig. 3d). We then tested whether the distribution pattern of transfected Six6os1 in COS7 cells was altered by its co-transfection with Syce1, Syce3, Syce2 or Tex12. This revealed that the SIX6OS1 distribution is drastically affected only in the presence of SYCE1 (from diffuse pattern to punctate, Fig. 3d; Supplementary Fig. 6b). Moreover, when COS7 cells were transfected with cDNAs encoding SYCP1, SYCE1, SYCE3 and SIX6OS1 simultaneously, all components co-localized in speckled cytoplasmic aggregates (Fig. 3e). This pattern was only altered when SYCE1 was absent (Supplementary Fig. 6c–f). We further validated the interaction between SYCE1 and SIX6OS1 in transfected COS7 cells by proximity ligation assay (PLA) (Supplementary Fig. 7). In summary, these results further support the findings of the Y2H and co-IP experiments by showing that SIX6OS1 interacts specifically and exclusively with SYCE1 in transfected COS7 cells.

**SIX6OS1 loading is dependent on synapsis.** To investigate the possible dependence of SIX6OS1 localization on the presence of other SC proteins, we analysed spermatocytes of mice deficient for Syce3 (ref. 30), Sycp1 (ref. 31) and meiotic cohesins Rad21l (ref. 4), Rec8 (ref. 32) and Stag3 (ref. 33). These meiotic mutants display different synaptic defects, from mild to more severe. In the absence of RAD21L or REC8, double labelling of SIX6OS1 and SYCP3 shows that SIX6OS1 is localized to synapsed-like regions (Fig. 4a). Interestingly, in Rec8 mutants, where there is no synapsis between homologues but instead the AEs of 40 univalents are decorated with SYCP1 as a result of 'synapsis-like' events between sister chromatids[34], SIX6OS1 is also present at these atypical synapsed-like regions (Fig. 4b). In Stag3-deficient mice, in which spermatocytes show almost no synapsis and very short AEs, SIX6OS1 also mimics SYCP1 localization. Finally, in mice lacking the central element proteins SYCE3 and SYCP1, in which AEs completely fail to synapse in a pachytene-like stage[30,31], SIX6OS1 was not detected despite the presence of a weak discontinuous pattern of SYCP1 deposition in the Syce3 mutant (Fig. 4; Supplementary Fig. 9c)[30]. These results, obtained through immunofluorescence analysis, allow a precise comparison of the different CE-mutant phenotypes (compare Fig. 4 with Fig. 7a by Schramm et al.[30]), and thus provide a global picture of the biology of the CE proteins. In this regard, we predict that Syce1 mutants will also be defective in SIX6OS1 loading since SYCE3 deficiency leads to failure in loading of both SYCE1 (ref. 30) and SIX6OS1 onto the LEs (Fig. 4).

Together, our results indicate that SIX6OS1 is a new protein of the CE, and its loading is consequently dependent on the assembly of the tripartite SC structure that occurs upon synapsis between homologous chromosomes or, interestingly, even between sister chromatids.

**Mice lacking SIX6OS1 are infertile.** To investigate the function of SIX6OS1 we generated a mutation of the murine Six6os1 gene by CRISPR/Cas9 genome editing (Fig. 5a). The a priori most suitable null mutation was chosen by PCR sequencing of the targeted region of the murine Six6os1 gene (Fig. 5b). A founder line was crossed with wild-type C57BL/6J and the resulting heterozygotes were interbred. Spermatocytes from homozygous targeted mice showed no SIX6OS1 protein expression by immunofluorescence when analysed using two independent polyclonal antibodies (Fig. 5c; Supplementary Fig. 3c). These results indicate that the mutation is a null allele of the Six6os1 gene (herein Six6os1$^{-/-}$).

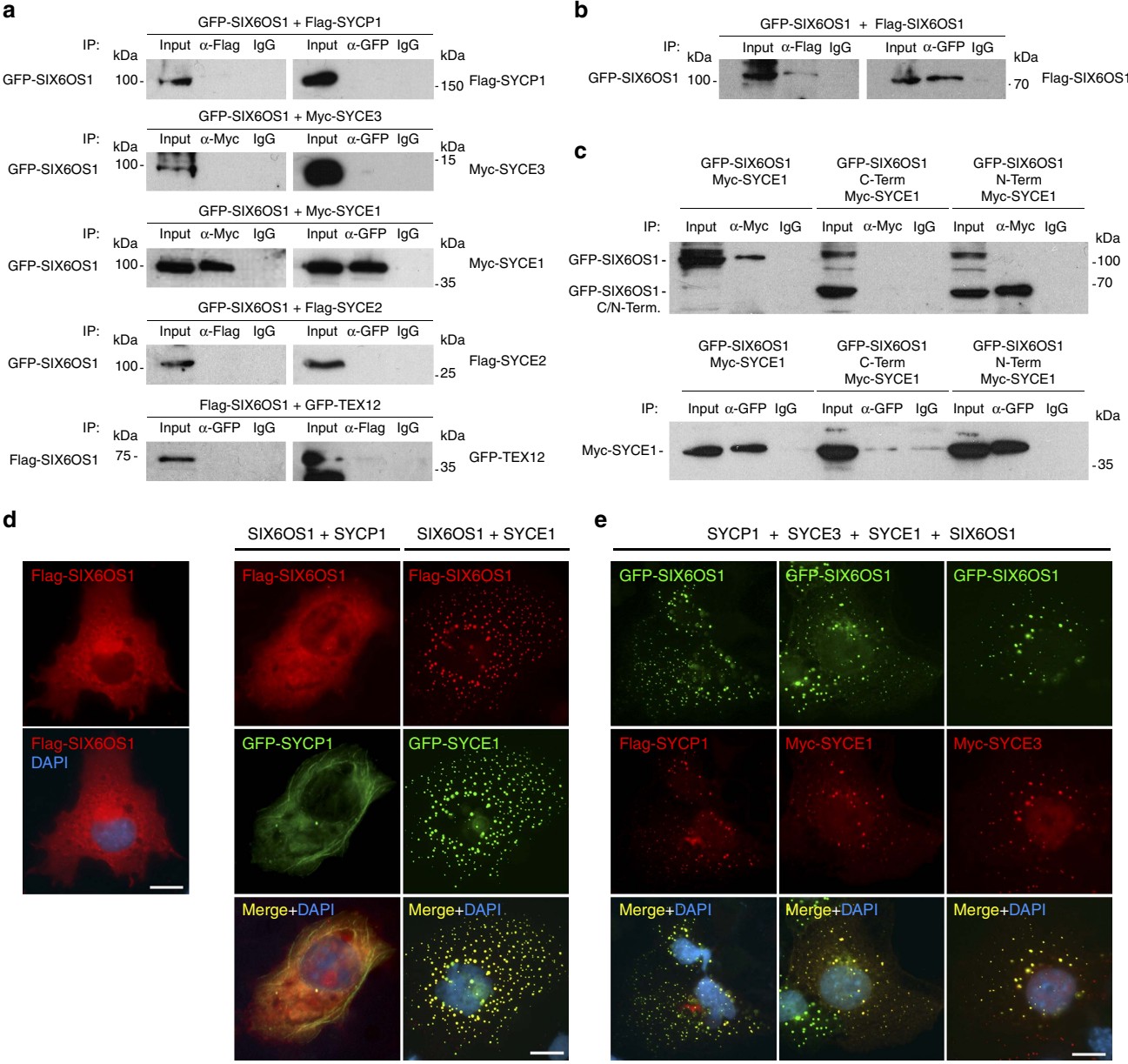

**Figure 3 | SIX6OS1 interacts specifically with SYCE1.** (**a–c**) HEK 293T cells were transfected or co-transfected with the indicated expression vectors. Protein complexes were immunoprecipitated overnight with either an anti-Flag, anti-EGFP or anti-Myc antibody, and were analysed by immunoblotting with the indicated antibody. (**a**) SIX6OS1 co-immunoprecipitates (co-IP) with SYCE1 (as well as in the reciprocal IP) but not with either SYCP1, SYCE3, SYCE2 or TEX12. (**b**) SIX6OS1-Flag co-immunoprecipitates with SIX6OS1-GFP, suggesting that it is able to form at least dimers. (**c**) SYCE1 co-immunoprecipitates with the SIX6OS1 N-terminal half (1–286) but not with the C-terminal half (287–574). Immunoprecipitation of SYCE1 and full length SIX6OS1 was used as positive control. (**d**) COS7 cells were transfected with *Six6os1* alone (left panel) or in combination with *Sycp1* and *Syce1* (right panel). SIX6OS1 localization drastically changed in the presence of SYCE1 but not with SYCP1. (**e**) *Sycp1*, *Syce3*, *Syce1* and *Six6os1* were simultaneously co-transfected in COS7 cells and found to co-localize in the cytoplasm in the punctate pattern of SYCE1. The experiments were reproduced three times. Scale bars, 15 μm.

Mice lacking SIX6OS1 did not display any obvious abnormalities but were sterile. Consistent with this, testes size from *Six6os1*$^{-/-}$ mice was only 30% of wild-type mice, and their epididymides exhibited complete absence of spermatozoa (Fig. 6a,b). Histological analysis of adult *Six6os1*$^{-/-}$ testes revealed seminiferous tubules that lacked postmeiotic cell types. The presence of spermatogonia, spermatocytes and Sertoli and Leydig cells was not altered. (Fig. 6b). By identifying groups of associated germ cell types in seminiferous tubule sections, the twelve stages of the epithelial cycle can be distinguished. Following these criteria, mutant adult mice showed an arrest at stage IV of the epithelial cycle. Spermatogenesis proceeds apparently normally in these mice up to prophase I, and then at stage IV, there is a massive apoptosis of spermatocytes (Fig. 6b). At 18 days of age, extensive apoptosis was also detected (Fig. 6c), suggesting that SIX6OS1 deficiency already affects spermatocytes during the first wave of meiosis. Thus, we conclude that SIX6OS1 is essential for spermatogenesis and its deficiency leads to non-obstructive azoospermia and consequently to infertility.

Histological analysis of whole ovaries of *Six6os1*$^{-/-}$ female mice at 4 months of age showed a lack of oocytes and a dense stroma (Fig. 6d). To investigate when this ovarian failure occurred, we histologically analysed ovaries from 6 day old

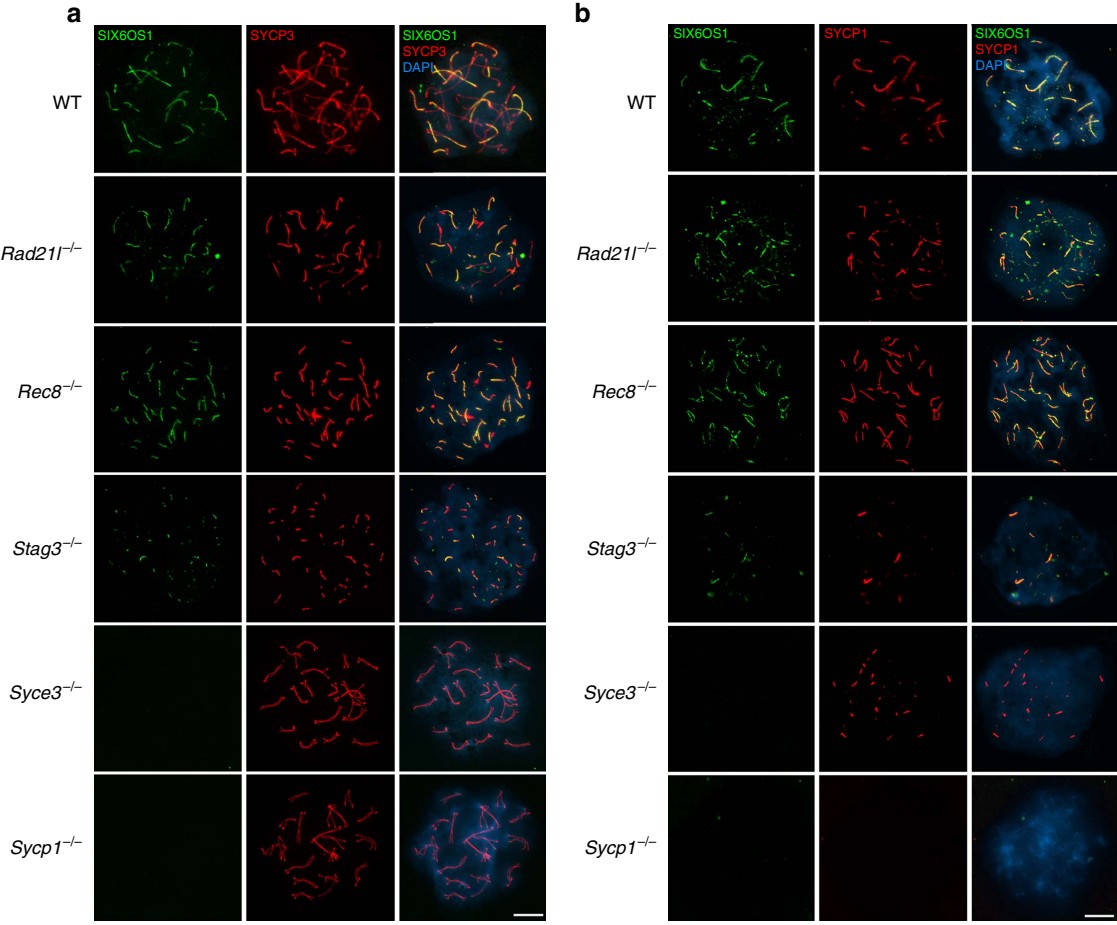

**Figure 4 | SIX6OS1 loading is dependent on synapsis but not on AE proteins.** (**a**) Double labelling of SIX6OS1 (green) and SYCP3 (red) or (**b**) SYCP1 (red) in $Rad21l^{-/-}$, $Rec8^{-/-}$, $Stag3^{-/-}$, $Syce3^{-/-}$ and $Sycp1^{-/-}$ showing that loading of SIX6OS1 is dependent on synapsis. SIX6OS1 is detected in the synapsed LEs of meiotic cohesin mutants but is absent from unsynapsed AEs in $Syce3^{-/-}$ and $Sycp1^{-/-}$ spermatocytes. Scale bar, 10 μm.

females (6 d.p.p.), a time point at which all oocytes are arrested in dictyate and present a large number of primordial follicles (outer cortex) and growing oocytes (inner cortex) (Fig. 6d). At 6 d.p.p., ovaries of $Six6os1^{-/-}$ mice are already depleted of follicles and show a severe ovarian dysgenesis (Fig. 6d) that is responsible for the absence of oogenesis and consequently for the severe premature ovarian failure.

**SIX6OS1 is essential for chromosome synapsis.** To characterize the meiotic defect in detail, $Six6os1$-deficient meiocytes were analysed using spread preparations from males as well as from fetal females. They were initially stained for AEs proteins (that is, SYCP3), revealing that mutant spermatocytes have AEs of normal morphology and composition (Fig. 7). Further, cohesins SMC3, REC8, STAG3, RAD21L and SMC1B are all present in AEs together with SYCP3 in $Six6os1$-deficient mice (Supplementary Fig. 8). As expected, in wild-type spermatocytes homologues were aligned in close juxtaposition during zygotene, and full synapsis was achieved at pachynema (Fig. 7a; Supplementary Fig. 9a). However, in both male and female $Six6os1$-deficient mice, synapsis failed to develop between homologues and all meiocytes were arrested in a pachytene-like stage, in most cases with their AEs properly aligned (A-type). However, a subset of meiocytes, more frequently observed in oocytes than in spermatocytes, showed poorly or even completely unaligned chromosome pairs (U-type; $17.6 \pm 3.7\%$ in spermatocytes; $n = 3$ and $79.06 \pm 18.9\%$ in oocytes; $n = 3$,

Fig. 7a,b; Supplementary Fig. 9a). The lack of synapsis, and the absence of breaks in unsynapsed AEs, were further analysed by counting the number of centromeres (ACA staining, 21 versus 40 in spermatocytes and 20 versus 40 in oocytes, Supplementary Fig. 10a) and telomeres (RAP1 marker, 40 versus 80, Supplementary Fig. 10b) in arrested meiocytes. This confirmed complete desynapsis but with the full integrity of AEs (all of the AEs have two RAP1 signals at their ends). Finally, and to refine the stage of the blockade, we immunolabelled $Six6os1^{-/-}$ spermatocytes with the mid pachytene-specific histone variant H1t (ref. 35). The positive staining for H1t (Supplementary Fig. 10c) indicates that arrested spermatocytes reach the mid-pachytene stage.

To gain further insight into the synaptic defects, we double immunolocalized SYCP3 and SYCP1. In contrast to other CE mutants such as $Syce3$, and even more so for $Syce2$ (ref. 36) and $Tex12$ (ref. 37), $Six6os1$-deficient spermatocytes have reduced levels of SYCP1 labelling (93.70% reduction in $Six6os1^{-/-}$ versus 54.36% reduction in $Syce3^{-/-}$). Mutant oocytes, however, show a slightly weaker reduction of SYCP1 staining (79.28% reduction, Fig. 7a,b; Supplementary Fig. 9c for quantification). We next double immunolocalized SYCP3 with SYCE1, SYCE3, SYCE2 and TEX12, revealing the absence of staining of all CE components in $Six6os1^{-/-}$ spermatocytes (Fig. 7c) and oocytes (Supplementary Fig. 11). Similarly, the regulatory cohesin subunit Sororin, which is located at the CE[27], is also lacking in $Six6os1$-deficient spermatocytes (Supplementary Fig. 8).

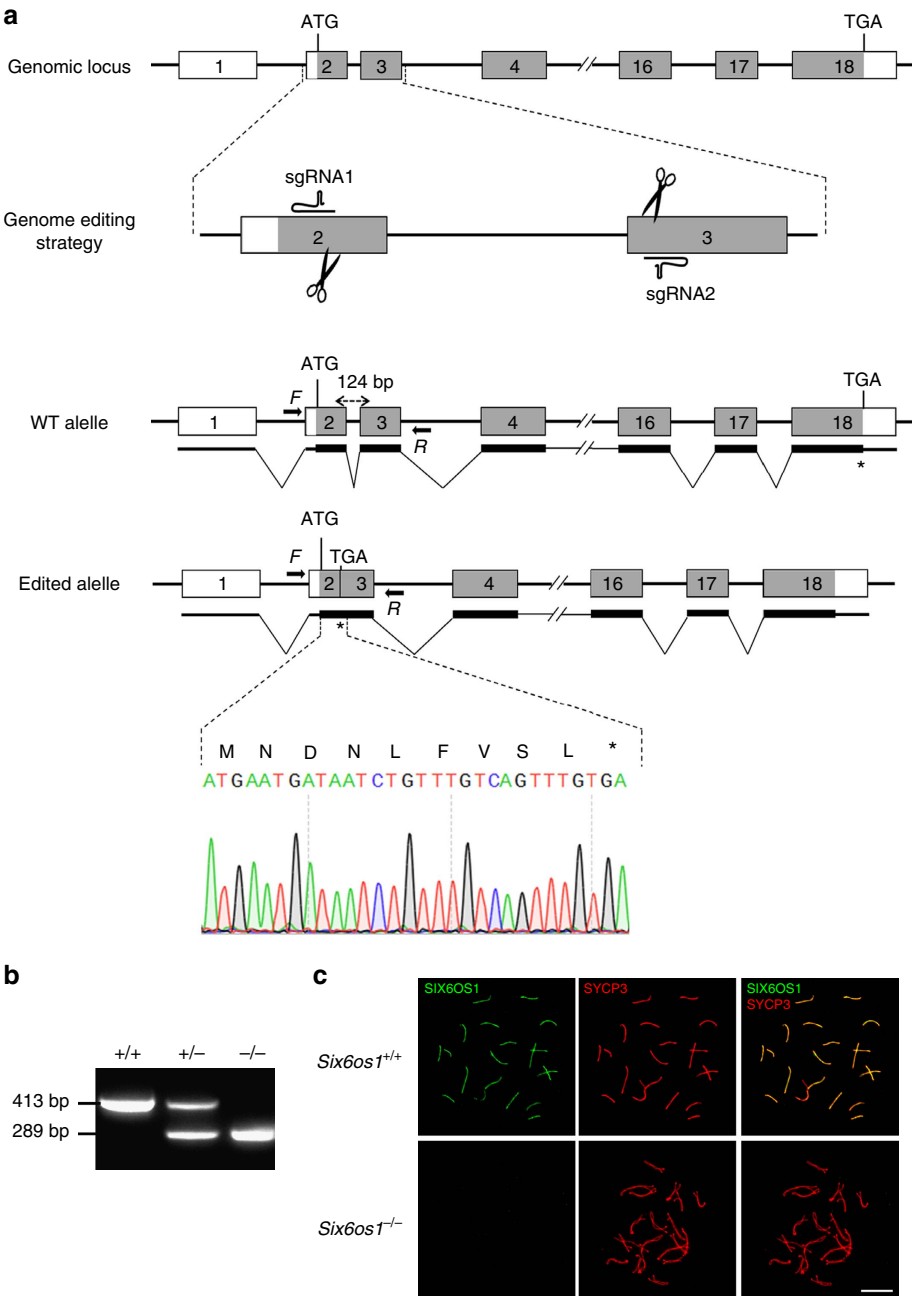

**Figure 5 | Generation and genetic characterization of *Six6os1*-deficient mice.** (**a**) Schematic representation of the wild-type locus (WT) and the genome editing strategy at the *Six6os1* locus, showing the sgRNAs, the corresponding coding exons (light grey) and non-coding exons (open boxes). Thin (non-coding) and thick (coding sequences) lines under exons represent the expected transcript derived from WT and *Six6os1* edited allele. ATG, initiation codon; TGA, stop codon. The nucleotide sequence of the 124 base pair deletion derived from PCR amplification of DNA from the *Six6os1* $^{edited/edited}$ is indicated. Primers are represented by arrows. (**b**) PCR analysis of genomic DNA from three littermate progeny of *Six6os1*$^{+/-}$ heterozygote crosses. The PCR amplification with primers F and R (indicated by arrows) revealed 413 and 289 bp fragments for wild-type and disrupted alleles, respectively. ($+/+$), ($+/-$) and ($-/-$) designate wild-type, heterozygous and homozygous knockout animals, respectively. (**c**) Double immunofluorescence of spermatocytes at pachytene stage obtained from *Six6os1*$^{+/+}$ and *Six6os1*$^{-/-}$ mice using SYCP3 (red) and a goat polyclonal antibody against SIX6OS1 (green). Scale bar, 10 μm.

To establish a direct causal role of SIX6OS1 deficiency in the observed phenotype, we analysed mice during the first almost synchronous wave of spermatogenesis at 18 d.p.p. We find that the meiotic arrest observed at this stage mimics that observed in adult males. Interestingly, the arrest is more homogeneous, with a lack of SYCP1 labelling in all AEs, despite the presence of both A- and U-type AEs (U-type $35.37 \pm 2.3\%$; Supplementary Fig. 7b). This suggests that the weak SYCP1 staining observed in the adult mutants could be a byproduct of a longer arrest.

In *Sycp1*-deficient mice, SIX6OS1 (Fig. 4), SYCE1-3 and TEX12 are not recruited to the SC[30]. Surprisingly, the lack of any of these central element proteins also leads to the aberrant deposition of SYCP1 in a weak discontinuous pattern, with the severest phenotype occurring in SIX6OS1 deficiency (weakest staining). This mutual interdependence, in addition to the fact that biochemical reactions (that is, DSB processing) take place in the three-dimensional (3D) mesh of the SC, make it difficult to distinguish cause and effect when analyzing mutant mice such as

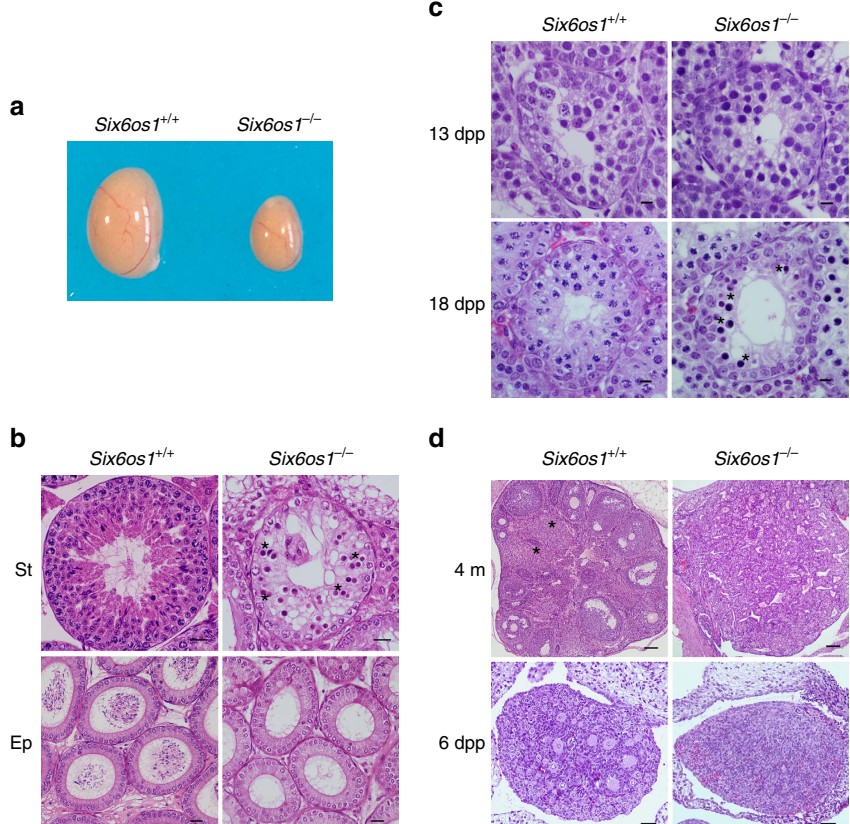

**Figure 6 | The absence of SIX6OS1 provokes azoospermia and ovarian failure.** (**a**) Genetic ablation of *Six6os1* leads to a reduction of the testis size ($n = 8$ wild-type and knock out, Welch's *t*-test analysis: $P < 0.0001$), and (**b**) a complete arrest of spermatogenesis in epithelial stage IV as shown in hematoxylin-eosin stained testis sections. Massive apoptosis of spermatocytes is indicated (asterisks). The spermatogenic arrest leads to empty epididymides and azoospermia. Scale bar in upper panels, 100 μm and in lower panels, 5 μm. (St) Seminiferous tubules. (Ep) Epididymides. (**c**) Tubule degeneration in juvenile mice (13 d.p.p. and 18 d.p.p.) lacking SIX6OS1 and spermatogenic arrest before pachytene studied by histology. At 13 d.p.p. spermatogenesis has reached late zygotene; at 18 d.p.p. it has reached late pachytene. Spermatocyte degeneration (apoptosis is indicated by asterisks) was first seen in 18 d.p.p. *Six6os1$^{-/-}$*. (**d**) Ovaries from *Six6os1*-deficient mice show atrophy with fibrosis and depletion of follicles. Comparative histological analysis of ovaries from *Six6os1$^{-/-}$* and wild-type mice at 6 days (6 d.p.p.), and 4 months (4 m) of age. Asterisks indicate corpora lutea. Scale bars represent 100 μm in 4 m, and 20 μm in 6 d.p.p.

those of the SC. Based on recent progress in elucidating the organization of the SC[38], and on the specific interaction of SIX6OS1 with SYCE1, it seems most plausible that SYCP1 recruits CE proteins, and the nascent CE then stabilizes SYCP1 assembly. Thus, absence of CE proteins disrupts the full accumulation of SYCP1, leading to weak/discontinuous staining patterns.

**Defective DSB processing in *Six6os1$^{-/-}$* meiocytes.** During leptonema, DSBs are generated by SPO11 and are then resected to form ssDNA ends that invade into the homologous chromosome. DSBs are marked by the presence of phosphorylated H2AX (γ-H2AX)[39], which is formed through phosphorylation by the kinase ATR following its recruitment by BRCA1 (ref. 37). Thus, we monitored the formation of DSBs by analyzing the presence of γ-H2AX. While γ-H2AX distribution in mutant spermatocytes resembles that of wild-type cells in early prophase I (leptotene, zygotene) (Supplementary Table 1), γ-H2AX is not restricted to sex chromosomes during pachynema (Fig. 8a). In contrast, γ-H2AX shows a moderate labelling on the chromatin of AEs in mutant pachytene-like spermatocytes (WT $23.40 \pm 3.2$; U-type $26.02 \pm 5.0$; A-type $30.74 \pm 3.6$; see Supplementary Table 2). In females, the distribution of γ-H2AX is slightly different. Oocytes at pachytene-like stage show a similar overall pattern of γ-H2AX labelling as spermatocytes (WT $20.85 \pm 3.5$; U-type

$28.14 \pm 9.8$; A-type $27.01 \pm 10.9$; see Supplementary Table 2), but it is more strictly localized to their AEs (Fig. 8b). The distribution of γ-H2AX-labelling during early prophase I, and its persistence in meiocytes during the pachynema-like stage, suggest that DSBs are generated in *Six6os1$^{-/-}$* meiocytes but are not properly repaired. To better understand the processing of DSBs, we explored the kinetics and distribution of proteins involved in DSB recombination and repair. After DSBs are induced, the recombinase RAD51 is recruited to early recombination nodules and promotes homologous strand invasion[40]. In wild-type zygotene spermatocytes, RAD51 assembles on numerous foci along the AEs/LEs, which are substituted by the single strand binding protein RPA and finally disappear towards pachytene, with the exception of the unsynapsed sex AEs (Fig. 8a). During early stages (leptonema), RAD51 distribution in mutants was similar to wild-type controls (Supplementary Table 1). However, in *Six6os1$^{-/-}$* spermatocytes at zygotene and pachytene-like stage, both RAD51 and RPA remained partially associated with the AEs (Fig. 8a; see Supplementary Tables 1 and 2). We obtained similar results when we analysed spreads from *Six6os1$^{-/-}$* oocytes (Fig. 8b; Supplementary Table 2).

Next, we analysed the presence of MSH4 and MLH1 foci in mutant spermatocytes. MSH4 mediates the transition after synapsis from initial to late recombination nodules. MLH1 is a component of the post-replicative mismatch repair system and

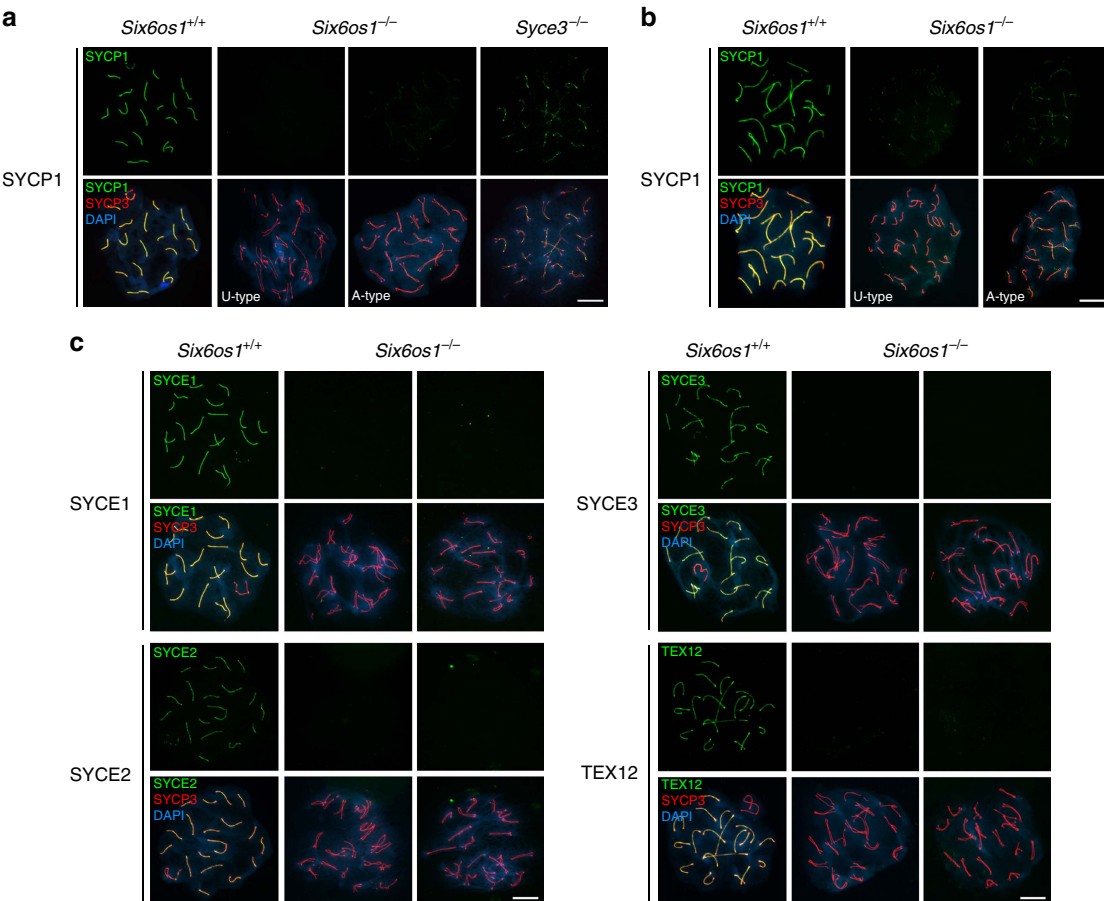

**Figure 7 | Six6os1⁻/⁻ meiocytes are not able to synapse. (a)** Double labelling of spermatocyte spreads of wild-type pachytene, and *Six6os1⁻/⁻* and *Syce3⁻/⁻* arrested pachytene-like spermatocytes with SYCP3 (red) and SYCP1 (green). In *Six6os1⁻/⁻* spermatocytes, SYCP1 does not localize to the unaligned-type (U-type) AEs but shows a very weak staining in spermatocytes with more aligned AEs (aligned-type, A-type). By direct comparison, in *Syce3⁻/⁻* arrested spermatocytes SYCP1 localizes in a discontinuous pattern along AEs independent of whether or not they are closely aligned. **(b)** Double labelling of spreads of wild-type pachytene and *Six6os1⁻/⁻* pachytene-like oocytes (aligned and unaligned) with SYCP3 and SYCP1. **(c)** Double labelling of spreads of wild-type pachytene and *Six6os1⁻/⁻* pachytene-like spermatocytes of SYCP3 (red) and SYCE1, SYCE3, SYCE2 or TEX12 (green) (see also extended Supplementary Fig. 11 for staining in oocyte spreads). All proteins are completely absent from the AEs in *Six6os1*-deficient mice. Scale bar, 10 µm.

marks sites of future chiasmata[41,42]. During early stages, MSH4 foci in mutants resemble those in wild-type spermatocytes. However, these foci persisted in pachytene-like arrested spermatocytes from *Six6os1*-deficient mice (Supplementary Fig. 12). Lastly, MLH1 foci were absent in *Six6os1⁻/⁻* pachytene-like chromosomes (Fig. 9a), while one/two MLH1 foci per bivalent were observed in wild-type spermatocytes. Similar results were obtained in oocytes lacking SIX6OS1 (Fig. 9a), suggesting a direct function of SIX6OS1 in homologous recombination rather than in the elimination of arrested spermatocytes at the so-called pachytene checkpoint of males. To further validate this, and in light of the late arrest at mid-pachytene-like stage (H1t positive), we exposed mutant spermatocytes to the PP2A inhibitor okadaic acid to allow *in vitro* transition from pachytene to metaphase-like I (ref. 43). After okadaic acid treatment, there was a rapid induction of chromosome condensation, leading to 20 bivalents that stain for SYCP3, with the formation of at least one chiasma in the wild type (Fig. 9b). In contrast, okadaic acid-treated *Six6os1⁻/⁻* spermatocytes displayed 40 free univalents, with characteristic labelling for SYCP3 (Fig. 9b). Together, our data strongly suggest that processing of recombination intermediates into MLH1-marked late recombination nodules (chiasmata) is critically dependent on SIX6OS1.

**X–Y chromosome behaviour and sex body formation.** In *Six6os1* mutant spermatocytes, the X and Y chromosomes are aligned in only $25.49 \pm 0.06\%$ of cells (Fig. 10a). In contrast, the degree of alignment of the sex bivalent in *Syce3* null mutants is $44.10 \pm 3.17\%$. In mutant spermatocytes that lack aligned sex chromosomes, the sex body is not formed (see below H2AX staining Fig. 10b). The remaining fraction of *Six6os1⁻/⁻* spermatocytes with aligned XY chromosomes show apparent synapsis at the PAR, but without staining for SYCP1, in contrast to the positive SYCP1 labelling of the PAR in *Syce3⁻/⁻* spermatocytes (Fig. 10a). These results suggest that whilst the sex body is not formed in either mutant, the synapsis defect in the absence of SIX6OS1 is more severe than in the *Syce3* knockout.

The X and Y chromosomes show homology only along the distal pseudoautosomal region[44] of their chromosome lengths, and the remaining unsynapsed parts are subjected to meiotic sex chromosome inactivation. This is a meiotic specific process that uses the DNA damage response to recognize unsynapsed regions and reconfigure their chromatin to a silent epigenetic domain named the sex body. The act of silencing is itself dependent upon phosphorylation of histone H2AX (γ-H2AX) by ATR in a BRCA1-dependent manner[45]. We performed γ-H2AX labelling of mutant spermatocytes and found moderate staining of the X and Y chromosomes in those cells showing aligned sex

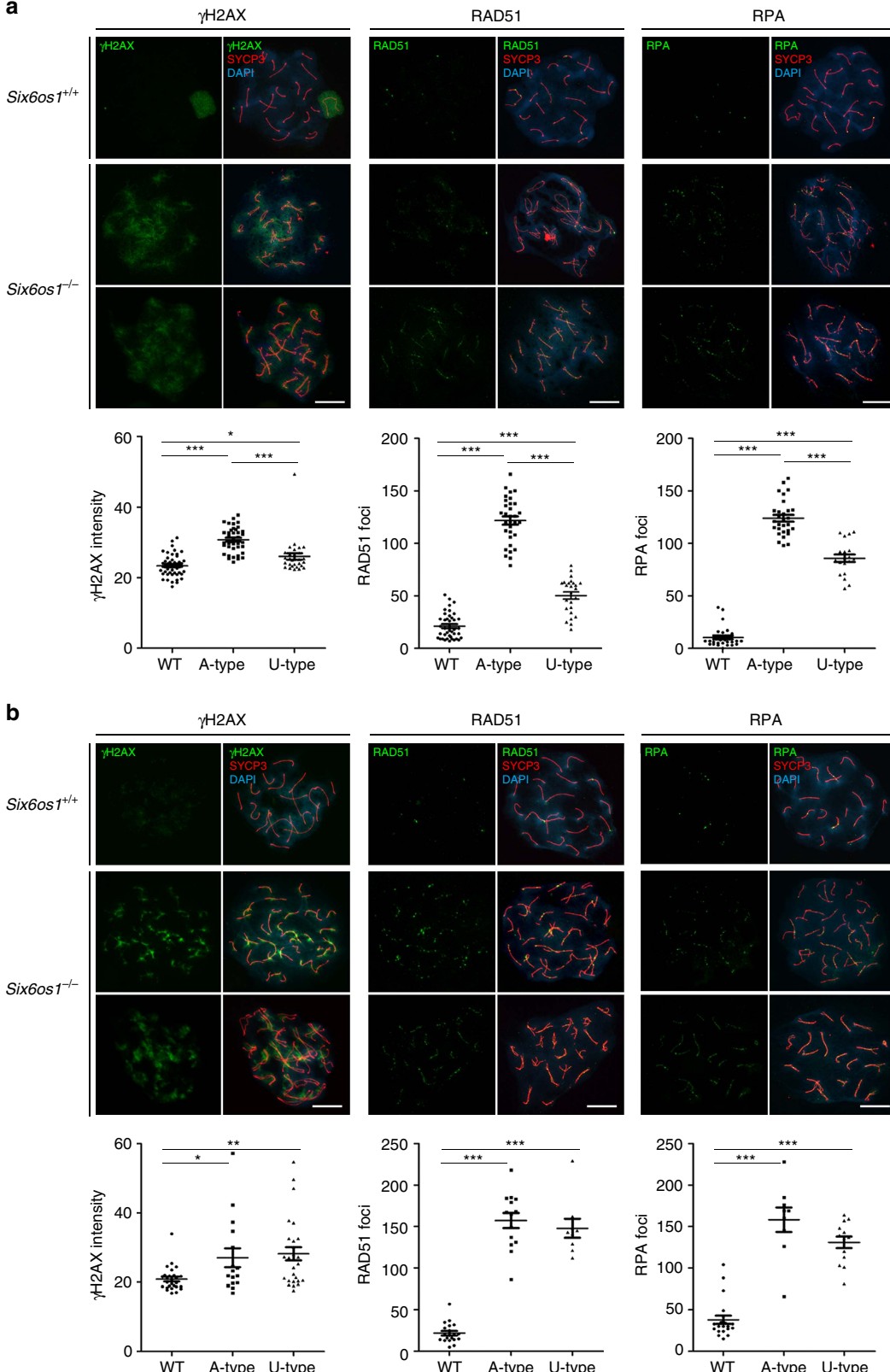

**Figure 8 | DSBs are generated but defectively repaired in *Six6os1*-deficient meiocytes.** (**a**) Double immunolabelling of γ-H2AX (green) with SYCP3 (red) in wild-type and *Six6os1$^{-/-}$* spermatocytes (left panel). In wild-type pachytene, γ-H2AX labels intensely the chromatin of the unsynapsed sex bivalent. In *Six6os1$^{-/-}$* pachytene-like spermatocytes γ-H2AX labelling remains in the chromatin. Double immunolabelling of SYCP3 (red) and RAD51 (green) (central panel) or RPA (green) (right panel). Both RAD51 and RPA remain associated to the AEs in *Six6os1$^{-/-}$* pachytene-like spermatocytes, showing a higher number of foci than wild-type pachytene. (**b**) Immunostaining of spread preparations of wild-type pachytene and *Six6os1$^{-/-}$* pachytene-like oocytes for γ-H2AX (green), RAD51 (green) and RPA (green) together with SYCP3 (red). γ-H2AX labelling in *Six6os1$^{-/-}$* arrested oocytes is more restricted to the AEs than in spermatocytes. Plots under each image panel represent the quantification of intensity or number of foci from wild-type and pachytene-like arrested meiocytes. Welch's *t*-test analysis: \*P < 0.01; \*\*P < 0.001; \*\*\*P < 0.0001. (See numeric data at Supplementary Table 2). Scale bar, 10 μm.

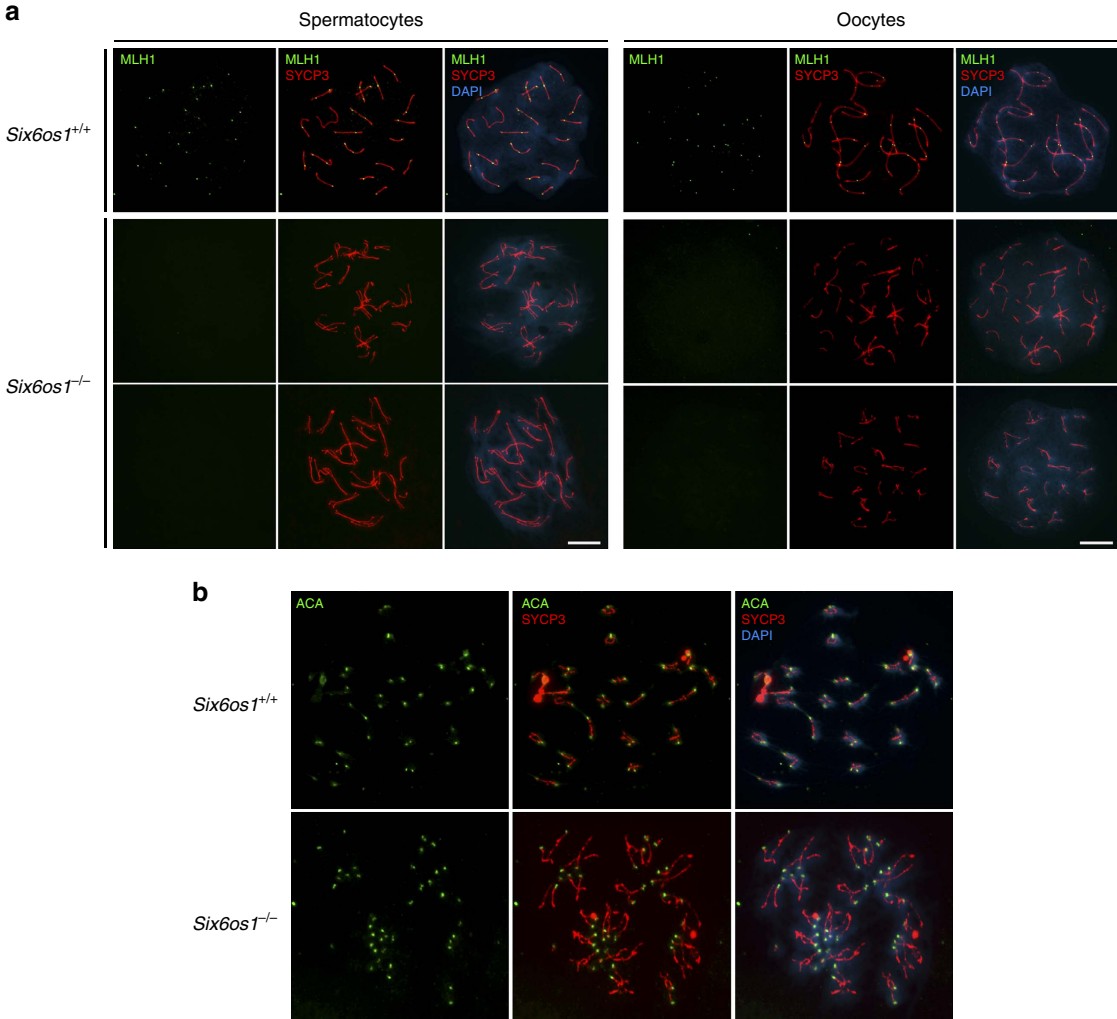

**Figure 9 | SIX6OS1 is essential for the processing of intermediate recombination nodules.** (**a**) Double immunolabelling of SYCP3 (red) with MLH1 (green). MLH1 foci are absent at the AEs/LEs of $Six6os1^{-/-}$ meiocytes whereas at least one focus is present along each autosomal SC in wild-type pachytene meiocytes. (**b**) Immunostaining of SYCP3 (red) and ACA (green) in wild type and $Six6os1^{-/-}$ spermatocytes. Okadaic acid-induced metaphase I plates of wild-type spermatocytes give rise to 20 bivalents each, with two opposed centromere signals (ACA) and positive staining for SYCP3, whereas $Six6os1^{-/-}$ spermatocytes lead to 40 free univalents, each with an ACA signal and SYCP3 labelling the centromeric and interchromatid domain. Scale bar, 10 μm.

chromosomes. This is in contrast with the strong labelling observed in the sex body chromatin of control spermatocytes at pachynema (Fig. 10b). Given the interplay between synapsis and DNA damage response, we directly analysed the status of 53BP1, a component of the DNA damage response that collaborates with BRCA1 in the sex body formation. In contrast to its accumulation on the unsynapsed XY[45] in wild type, 53BP1 signals were not observed in $Six6os1$ mutant spermatocytes (Fig. 10c). Together, these results indicate that SIX6OS1 deficiency, similar to most asynaptic mice mutants, impedes sex body formation[46].

## Discussion
Synapsis of homologous chromosomes is essential for the completion of meiosis and thus for fertility. The SC provides the structural framework for synapsis and for the processing of recombination intermediates into crossovers. Recently, the gene variant rs1254319 (p.Leu524Phe) in the anonymous $C14ORF39/$ $SIX6OS1$ gene was identified as an influencing polymorphism affecting the human recombination rate[2]. In addition, the same rs1254319 (p.Leu524Phe) variant has been associated with age at menarche, an indirect fertility trait[47], in a meta-analysis of 32

genome-wide association studies in 87,802 women of European descent[48]. Accordingly, we show that this coding variant of $SIX6OS1$ lies in a conserved residue of the SIX6OS1protein. Cytological analysis revealed that SIX6OS1 is a new component of the CE of the SC that co-localizes with SYCE1 and SYCE3 at the synapsed chromosome axes (Fig. 1e).

By comparison of the cytological localization of CE proteins, their protein–protein interaction network, and the phenotypes obtained from their knockout mice, it has been suggested that there are two discernible subdomains within the central element. One domain, formed by SYCP1, SYCE1 and SYCE3, would act in concert through a network of interactions, specifically between SYCP1 and SYCE3, and between SYCE3 and SYCE1 (refs 49,50). The other, more inner domain of TEX12 and SYCE2 would form a separate complex by themselves. The SYCE2–TEX12 complex is an equimolar hetero-octamer, formed by the association of a SYCE2 tetramer and two TEX12 dimers[21], and their corresponding mutant mice show some degree of synapsis with small but intense foci of SYCP1, SYCE1 and SYCE3 between their aligned AEs[24,30,51]. Mutant spermatocytes for $Syce3$ show an intermediate phenotype, with defective recruitment of CE proteins but a weak discontinuous pattern of SYCP1

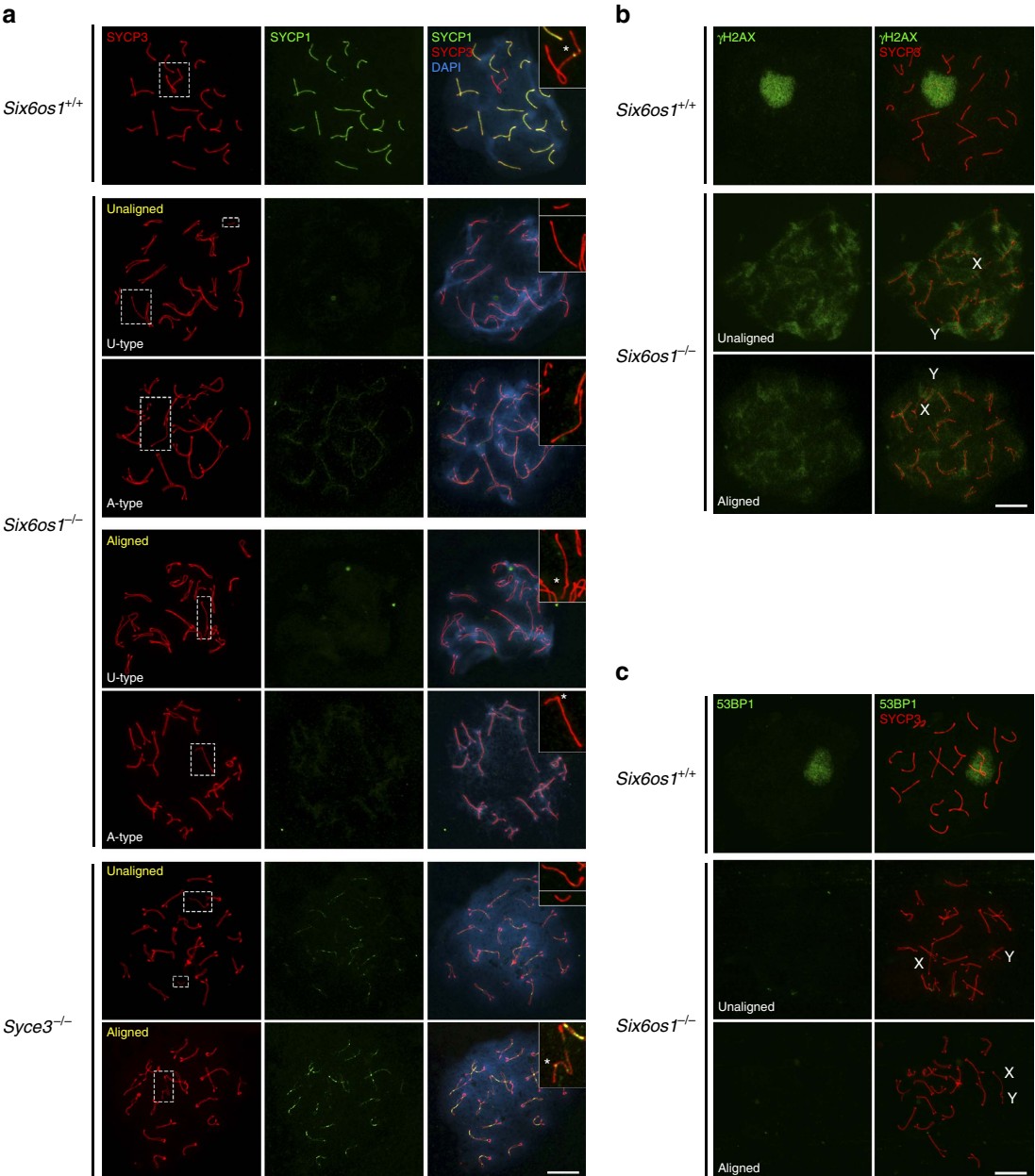

**Figure 10 | X–Y chromosome behaviour.** (**a**) Double immunolabelling of SYCP3 (red) and SYCP1 (green) in wild-type pachytene, and $Six6os1^{-/-}$ and $Syce3^{-/-}$ pachytene-like arrested spermatocytes. Yellow letters indicate aligned/unaligned sex chromosomes in mutant spermatocytes. (**b**) Co-labelling of SYCP3 (red) and γ-H2AX (green) in $Six6os1^{-/-}$ pachytene-like arrested spermatocytes, showing moderate staining in XY chromosomes. (**c**) Double immunolabelling of SYCP3 (red) and 53BP1 (green). 53BP1 signal is not observed in the XY chromosomes of mutant spermatocytes, in contrast to the strong labelling of the sex body in wild type. Scale bar, 10 μm.

staining[30]. SIX6OS1 deficiency produces a more severe phenotype, with weaker discontinuous SYCP1 loading and also lack of recruitment of CE proteins (Fig. 7). Thus, SIX6OS1 would belong to the first subdomain (together with SYCP1, SYCE1 and SYCE3), in which mutant mice display aligned homologues and normally assembled AEs, but no CE structures at all. From a cytological perspective, the pattern of SIX6OS1 distribution along LEs in pachynema (revealed through co-localization curves and regression coefficients) is also in agreement with the continuous distribution of SYCE1 and SYCP1 along synapsed LEs[24,30] (Fig. 2). This localization of SIX6OS1 is also congruent with the Y2H, co-immunoprecipitation and PLA results we have obtained showing its specific ability to interact and co-localize with SYCE1. Taken together with the presence of more unaligned chromosome pairs in $Six6os1^{-/-}$ than in $Syce3^{-/-}$ spermatocytes, and the interdependent loading of SIX6OS1 and SYCE3 (Figs 4a,b and 7c), we suggest that SIX6OS1 is required at a similar hierarchy level (neither upstream nor downstream) to SYCE3, and downstream of SYCP1.

Despite the recent advances in reconstructing the 3D molecular organization of the mammalian SC with isotropic resolution through super-resolution imaging[52], several gaps still remain in the net of interactors and partners involved in the assembly of the SC tripartite structure. In this regard, it has been postulated that SYCE1 stabilizes SYCP1 N-terminal interactions in the CE[25], suggesting that SYCE1 can act alongside SYCP1 with SYCE3. However, this view is neither validated by the phenotype of $Syce3^{-/-}$ spermatocytes, in which SYCP1 stains weakly and

SYCE2/TEX12 is absent from AEs[30], nor by Y2H studies[21]. Recently, a direct interaction between SYCP1 and SYCE3, but not with SYCE1, SYCE2 or TEX12, has been shown by co-IP experiments and biochemical studies[50], which is in closer agreement with the genetic depletion phenotype. Similarly, with the present knowledge of interactions and components of the SC, we have no explanation for the weaker SYCP1 staining in adult *Six6os1* mutants in comparison with the *Syce3* mutant (which also lacks SIX6OS1). In this sense, the appearance of a new player in the CE family of proteins such as SIX6OS1, which is essential for the stabilization of the central region and for synapsis, deepens the complexity of the multilayered structure of the CE and suggests that unknown players could help to elucidate several open questions.

It has been shown through mouse mutants lacking CE-specific proteins that assembly of the SC central region is essential for recombination progression and chiasmata formation[9]. Similarly, *Six6os1*-deficient meiocytes showed an arrest in the processing of recombination intermediates into MLH1-marked late recombination nodules (chiasmata). Together, these observations raise the possibility that interaction between components of the CE and recombination machinery would be critical for meiotic recombination. In this context, interactions have been described between RAD51 and both SYCP1 and SYCE2 (ref. 26). Based on sequence analysis, we predict that SIX6OS1 contains a highly helical structural region within its N terminus, followed by a flexible linker and then a C-terminal flexible protein docking sequence that could recruit multiple globular proteins to induce macromolecular protein complex. In accordance with other SC proteins, self-association and SYCE1-binding are likely mediated by the helical N-terminal domain, suggesting that this region may function in the structural assembly of the CE. The predicted unstructured nature of the remainder of the sequence suggests it may act as a flexible linker between the N-terminal structural domain and protein–protein interactions mediated by conserved patches of residues within the C terminus of the molecule. It is tempting to speculate that the predicted protein–protein interaction motifs of the C-terminal region may be responsible for the recruitment and/or stabilization of components of recombination nodules necessary for proper recombination progression. Consequently, subtle variations in the protein sequence of human SIX6OS1 (that is, rs1254319, p.Leu524Phe), could act by modifying the CO/NCO ratio, which is ultimately responsible for the observed number of recombination events genome wide. Interestingly, allele A in rs1254319 p.Leu524Phe is associated with higher recombination rate in women (53 cM) but not in men[2]. This observation fits well with the observed sexual dimorphism in several cellular aspects of *Six6os1*-deficient mice, such as differences in the deposition of SYCP1 (Fig. 7a,b; Supplementary Fig. 9c) and difference in the frequency of U-type AEs between mutant oocytes and spermatocytes.

In summary, we have identified the biological pathway by which the SNV identified in *SIX6OS1* affects the recombination rate in humans. Our functional data show how this protein of the SC is dispensable for the generation of DSBs, but is required for the appropriate processing of intermediate recombination nodules immediately before reciprocal recombination and CO formation, and is thus essential for chromosome synapsis and fertility.

## Methods

**Histology.** For histological analysis of adult testes, mice were perfused and their testes/ovaries were processed into serial paraffin sections and stained with hematoxylin-eosin. For histological studies of 13 and 18 day mice, testes were fixed in Bouin's fixative.

**Immunocytology and antibodies.** Testes were detunicated and processed for spreading using the 'dry-down' technique. Oocytes from fetal ovaries (E17.5 embryos) were digested with collagenase, incubated in hypotonic buffer, disaggregated, fixed in paraformaldehyde and incubated with the indicated antibodies for immunofluorescence. Goat polyclonal antibodies against C14ORF39/SIX6OS1 were developed by Santa Cruz (sc-245304) and used for the immunofluorescence analysis. This antibody was raised against a conserved internal region of human SIX6OS1. Rabbit polyclonal antibodies against SIX6OS1 were developed by Proteintech (22664-1-AP) against a fusion protein of GST with SIX6OS1 (C-350 aa) of human origin (see Supplementary Fig. 3 for validation) and was used to validate the immunofluorescence results obtained with the goat polyclonal antibody against C14ORF39/SIX6OS1 developed by Santa Cruz. The primary antibodies used for immunofluorescence were rabbit αSMC3 serum K987 (1:20), rabbit αSMC1β serum K974 (1:20), rabbit αSTAG3 serum K403, αREC8 serum K1019, rabbit αRAD21 IgG K854 (1:5)[4,5], mouse αSYCP3 IgG sc-74569 (1:100), rabbit αRAD51 sc-8349 (1:30) and PC130 (1:50), rabbit αSYCP1 IgG ab15090 (1:200) (Abcam), rabbit anti-γH2AX (ser139) IgG #07-164 (1:200) (Millipore), ACA or purified human α-centromere proteins IgG 15–235 (1:5, Antibodies Incorporated), mouse αMLH1 51-1327GR (1:5, BD Biosciences), rabbit α53BP1 sc-22760 (1:20), rabbit αRAP1 IgG (1:400, provided by Dr Titia de Lange, The Rockefeller University, USA), and rabbit αRPA IgG (1:300, provided by Dr E. Marcon, Toronto University, Canada), rabbit αTEX12 IgG (1:100) and guinea pig αSYCE3(1:20) (provided by Dr R. Benavente, University of Würzburg, Germany), guinea pig αSYCE1 (1:100), rabbit αSYCE1 (Proteintech), guinea pig αSYCE2 (1:50) (provided by C. Höög, Karolinska Institutet, Sweden) and guinea pig αH1t (Provided by MA Handel). The secondary antibodies used were TRITC α-mouse 115-095-146/α-rabbit 111-025-144 and FITC α-mouse 115-095-146/α-rabbit 111-095-045 (Jackson ImmunoResearch) (all 1:100). Slides were visualized at room temperature using a microscope (Axioplan 2; Carl Zeiss, Inc.) with 63 × objectives with an aperture of 1.4 (Carl Zeiss, Inc.). Images were taken with a digital camera (ORCA-ER; Hamamatsu) and processed with OPENLAB 4.0.3 and Photoshop (Adobe). Quantification of γH2AX and SYCP1 fluorescence signals was performed using Image J software. Chromosome counts of A-type and U-type cells were performed on at least 100 pachytene-like spermatocytes and oocytes from three individuals.

***In vivo* electroporation.** Testes were freed from the abdominal cavity and 10 μl of DNA solution (50 μg) mixed with 1 μl of 10 × FastGreen (Sigma Aldrich F7258) was injected in the rete testis with a DNA embryo microinjection tip. After a period of 1 h following the injection, testes were held between a pair of electrodes and electric pulses were applied four times (35 V for 50 ms each pulse) using a CUY21 BEX electroporator[23].

**Electron microscopy.** For immunoelectron microscopy, 10 μm cryosections of mouse testis were fixed with acetone for 10 min at − 20 °C and air dried. Incubation with primary antibodies was carried out in a humidified box for 4 h at room temperature. After rinsing twice in PBS, sections were fixed for 10 min in 2% formaldehyde and blocked with 50 mM NH₄Cl. Secondary antibodies conjugated to 6 nm gold particles were incubated overnight at 4 °C, and samples were subsequently washed in PBS. Samples were fixed for 30 min in 2.5% glutaraldehyde and postfixed in 2% osmium tetroxide. After rinsing three times with $H_2O$, samples were dehydrated in an ethanol series and embedded in Epon. Ultrathin sections were stained with uranyl acetate and lead citrate according to standard procedures[30].

**Okadaic acid assay.** Testes were dissected into a Petri dish containing ice cold sterile medium (4 mM L-glutamine, 10% fetal calf serum, and 25 mM Hepes in Dulbecco's Modified Eagle's medium) and cell suspensions ($5 \times 10^6$ cells per ml) were exposed to 5 μM okadaic acid (Sigma-Aldrich) for 5 h at 32 °C and 5% $CO_2$ before spreading the cells by the dry down procedure[4].

**Generation of plasmids.** Full-length cDNAs encoding SIX6OS1, TEX12, SYCE1, SYCE2, SYCE3, SYCP1 and SYCP3 were RT–PCR amplified from murine testis RNA. Full-length cDNAs were cloned into the pcDNA3, pcDNA3 x2Flag, pCEFL HA or pcDNA3.1 Myc-His (-) or pEGFP-C1 mammalian expression vectors.

**Cell lines and transfections.** HEK 293T and COS7 cell lines were transfected with Lipofectamine (Invitrogen) or Jetpei (PolyPlus) and obtained from the ATCC. Cell lines were tested for mycoplasma contamination (Mycoplasma PCR ELISA, Sigma).

**Immunoprecipitation and proximity ligation assay.** HEK 293T cells were transiently transfected and whole cell extracts were prepared and cleared with protein G Sepharose beads (GE Healthcare) for 1 h. The antibody was added for 2 h and immunocomplexes were isolated by adsorption to protein G Sepharose beads for 1 h. After washing, beads were loaded onto reducing 10% polyacrylamide SDS gels and proteins were detected by western blotting with the indicated antibodies. Immunoprecipitations were performed using mouse αFlag IgG (5 μg; F1804, Sigma-Aldrich), rabbit αMyc Tag IgG (4 μg; #06-549, Millipore), mouse αHA.11

IgG MMS- (5 µl, aprox. 10 µg per 1 mg prot; 101R, Covance), goat αGFP IgG (4 µg; sc-5385, Santa Cruz), ChromPure mouse IgG (5 µg/1 mg prot; 015-000-003), ChomPure rabbit IgG (5 µg per 1 mg prot.; 011-000-003, Jackson ImmunoR-esearch), ChomPure goat IgG (5 µg per 1 mg prot.; 005-000-003, Jackson ImmunoResearch). Primary antibodies used for western blotting were mouse αFlag IgG (F1804, Sigma-Aldrich) (1:10,000), rabbit αHA IgG (H6908, Sigma-Aldrich) (1:1,000), rabbit αFlag IgG (1:800; F7425 Sigma-Aldrich), mouse αMyc obtained from hybridoma cell myc-1-9E10.2 ATCC (1:1,000). Secondary horseradish peroxidase-conjugated α-mouse (NA931V, GE Healthcare), α-rabbit (#7074, Cell Signaling), or α-goat (A27014, Thermo Scientific) antibodies were used at 1:10,000, 1:3,000 or 1:10,000 dilution, respectively. Antibodies were detected by using Immobilon Western Chemiluminescent HRP Substrate from Millipore. Proximity Ligation Assay was performed using goat αSIX6OS1 (sc-5385) and rabbit αSYCE1, with the corresponding anti-goat PLA Probe PLUS and anti-rabbit PLA probe MINUS, following the manufacturer's instructions (Duolink Using PLA Technology, SIGMA).

The uncropped versions of western blots in Fig. 3 are shown in Supplementary Fig. 13.

**Production of CRISPR/Cas9-edited mice.** Six6os1-gRNAs (G68 5′-CACCGAT CTGTTTGTCAGTTTGGAC-3′ and 5′-AAACGTCCAAACTGACAAAC AG ATC-3′ and G75 5′-CACCGTACTTATGTCTT GCTCATAC-3′ and 5′-AAAC GTATGACAAGACATAAGTAC-3′ targeting exon 2 and exon 3 were predicted at crispr.mit.edu. Six6os1-sgRNAs were produced by cloning annealed complementary oligos at the BbsI site of pX330 (#42230, Addgene), generating PCR products containing a T7 promoter sequence that were purified (NZYtech), and then performing in vitro transcription using the MEGAshortscript T7 Transcription Kit (Life Technologies). The plasmid pST1374-NLS-flag-linker-Cas9 (#44758; Addgene) was used for generating Cas9 mRNA after linearization with AgeI. In vitro transcription and capping were performed using the mMESSAGE mMACHINE T7 Transcription Kit (AM1345; Life Technologies). Products were purified using the RNeasy Mini Kit (Qiagen). RNA (100 ng µl$^{-1}$ Cas9 and 50 ng µl$^{-1}$ each guide RNA) was microinjected into zygotes (F1 hybrids between strains C57BL/6J and CBA/J)[53]. Edited founders were identified by PCR amplification (Taq polymerase, NZYtech) with primers flanking exons 2 and 3 (Primer F 5′-CACTTACATTTTCCTTTTAAGAATGC-3′ and R 5′-CCCCTC TCAT ACATACAAGTTGC-3′) and subcloned into pBlueScript (Stratagene) followed by standard Sanger sequencing. The length of the corresponding wild-type and mutant allele were 413 and 289 bp, respectively. The selected founder was crossed with wild-type C57BL/6J to eliminate possible unwanted off-targets and to generate pure heterozygous. Six6os1$^{+/-}$ heterozygous mice were sequenced again by Sanger sequencing and crossed to give rise to Six6os1$^{-/-}$ homozygous. Genotyping was performed by agarose gel electrophoresis analysis of PCR products produced from DNA isolated from tail biopsy specimens. Mouse mutants for Rec8, Rad21l, Syce3, Sycp1 and Stag3 have been previously developed[4,13,30–32].

Mice were housed in a temperature-controlled facility (specific pathogen free, spf) using individually ventilated cages, standard diet and a 12h light-dark cycle, according to European Union regulations at the 'Servicio de Experimentación Animal, SEA'. Mouse protocols were approved by the Ethics Committee for Animal Experimentation of the University of Salamanca (USAL). We made every effort to minimize suffering and to improve animal welfare. Blinded experiments were not possible since the phenotype was very obvious between wild-type and Six6os1-deficient mouse for all of the experimental procedures used. No randomization methods were applied since the animals were not divided in groups/treatments. The minimum size used for each analysis was three animals/genotype. The mice analysed were between 2 and 4 months of age, except in those experiments where is indicated.

**Quantitative PCR.** Total RNA was isolated from various tissues of wild-type adult mice. To analyse the expression of Six6os1 and Rad21l mRNAs, equal amounts of cDNA were synthesized using SuperScript II Reverse Transcriptase (Invitrogen, Life Technologies) and Oligo (dT). qPCR was performed using FastStart Universal SYBR Green Master Mix (ROX) (Roche) and specific forward and reverse primers: qSIX6OS1_F 5′- GCTGAATGTGGAGATAAAGAG-3′ and qSIX6OS1_R 5′-AG GAGTTTCAGGAGTTTGAGG-3′; qRAD21L_F 5′-TTGCAGCTCACTGGGAG AAGA-3′ and qRAD21L_R5′-AGTCCTGGGCGAAATGTCATC-3′. All qPCR reactions were performed at 95 °C for 10 min, and then 40 cycles of 95 °C for 15 s and 62 °C for 1 min on the iQ5 Thermal Cycler (Bio-Rad). β-Actin was amplified as a housekeeping gene with the primers qβ-actin_F 5′-GGCACCACACCTTCT ACAATG-3′and qβ-actin_R 5′-GTGGTGGTGAAGCTGTAGCC -3′.

**Y2H assay and screening.** Y2H assay was performed using the Matchmaker Gold Yeast Two-Hybrid System (Clontech) according to the manufacturers' instructions. Mouse Six6os1 cDNA encoding the N terminus (1-138) was subcloned into the vector pGBKT7 and was used as bait to screen a mouse testis Mate & Plate cDNA library (Clontech Laboratories Inc.). Positive clones were initially identified on double dropout SD (synthetic dropout)–Leu/–Trp/X-α-Gal/Aureobasidin A plates before further selection on higher stringency quadruple dropout SD/–Ade/–His/–Leu/–Trp/X-α-Gal/Aureobasidin A plates. Pray plasmids were extracted from the candidate yeast clones and transformed into Escherichia coli. The plasmids from two independent bacteria colonies were independently grown, extracted and Sanger sequenced. Southern blotting was also used for plasmid screening.

**Sequence analysis.** Protein sequences were extracted from the UniProt database and analysed using Jalview 2 (ref. 54). Multiple sequence alignments and secondary structure predictions were performed using MUSCLE (EBI)[55] and Jpred 4 (ref. 56), respectively.

**Co-localization profile.** SIX6OS1 and either SYCP1, SYCE1, SYCE3, SYCE2 or TEX12 were stained on spreads of wild-type spermatocytes. Images were captured with identical camera settings. Fluorescence signals were measured along the 19 autosomal AEs of pachytene cells using the 'Plot profile' tool of ImageJ. Signal intensities were standardized, acquiring values between − 1 and 1, and the overlay profiles of SIX6OS1 and other CE proteins were plotted. Regression analysis for each pair of proteins was performed to determine the correlation between their profiles. The values of the coefficients of determination $R^2$ are shown in the scatter plots.

**Statistics.** To compare counts between genotypes at different stages, we used the Welch's t-test (unequal variances t-test), which was appropriate as the count data were not highly skewed (that is, were reasonably approximated by a normal distribution) and in most cases showed unequal variance. Asterisks denote statistical significance: *P value < 0.01, **P value < 0.001 and ***P value< 0.0001.

**Data availability.** Genomic DNA sequences of H. sapiens (human, 317761), M. musculus (mouse, 75801) are available on GenBank (http://www.ncbi.nlm.nih.gov/genbank/). Amino acid sequences of H. sapiens (Q8N1H7), M. musculus (NP_083381), P. troglodytes (Chimp, H2Q8E6), S. charissii (Tasmnaina devil, G3WQS7), O. anatinus (Latypus, F6ZZ02), P. sinensis (Chinese turtle, K7GAG2), G. gallus (Chick, E1C952) and L. chalumnae (West india coelacanth, M3XIB0) were obtained from the UniProt database (http://www.uniprot.org/). All remaining data generated in this study are available in the Article and Supplementary Information files or available from the authors upon request from the authors.

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

## Acknowledgements

We wish to express our sincere thanks to Drs Lu, Höög, Titia de Lange, Schimenti and Handel for providing antibodies and reagents (mice), and to Dr JL de la Pompa for reviewing the MS and Dr Jessbrguer for useful help. This work was supported by BFU_2014-59307-R, MEIONet and JCyLe (CSI052U16). LGH and NFM are supported by European Social Fund/JCyLe grants (EDU/1083/2013 and EDU/310/2015). ORD is a Sir Henry Dale Fellow jointly funded by the Wellcome Trust and Royal Society (Grant Number 104158/Z/14/Z). RB is funded by DFG (grant Be1168/8-1). AT and ID were supported by DFG grants TO421/8-2 and TO421/6-1, respectively.

## Author contributions

L.G.H. and N.F.M. performed the characterization of the mutant mice including the cytological and biochemical analysis. M.S.M. carried out the Cas9 injections. O.R.D. carried out the protein analysis. I.R. carried out infertility phenotyping of mutant mice. I.G.T. contributed with the initial co-IP experiments. DdR performed the staging of the seminiferous tubules. J.L.B. contributed with reagents and discussion. R.B. performed the E.M. work and contributed with the Syce3 KO samples and discussion of the results. E.L.C. performed the Y2H analysis. I.D. and A.T. provided spermatocytes spreads from Sycp1 KO. A.M.P. and E.L.C. designed the experiments and wrote the paper with the input of the remaining authors.

## Additional information

Competing financial interests: The authors declare no competing financial interests.

