## [Peer Review File · Nature Communications]

Reviewer #1 (Remarks to the Author)

This manuscript presents a detailed analysis of the structure and phenotype of a novel gene implicated in controlling rate of human meiotic recombination, C14ORF39/SIX6OS1 (herein referred to as SIX6OS1, italicized, in human, and Six6os1, italicized, in mouse; and encoding the protein SIX6OS1, not italicized). The authors identified this gene by systematic analysis of previously published lists of coding variants of genes controlling the rate of meiotic recombination in an Icelandic population. Together, the findings of this paper establish that SIX6OS1 protein is a component of the central element (CE) of the synaptonemal complex (SC), where it co-localizes and interacts with SYCE1. Mice lacking functional Six6os1 are infertile, with defects in homologous chromosome synapsis and subsequent arrest of germ cells during meiotic prophase.

Overall, this is a very good report: the quality of the work is high, the detail is meticulous, and the conclusions are justified. However, attention to the following suggestions would improve the manuscript:

1. There is no overt validation of the antibody recognizing SIX6OS1, although Fig. 5c shows that it recognizes a 70 kD band and is absent in the mutant mice. The Materials and Methods should include a statement on how the antibody was developed and validated, and also a complete gel should be shown (does it recognize only a single band?).
2. The section "Sequence analysis of SIX6OS1" seems out of place and should be included within the first section of Results ("C14orf39/SIX6OS1 is a novel protein composing the CE of the mammalian SC").
3. Protein interaction studies are comprehensive. However, they are limited to in vitro interactions in transiently transfected HEK293 cells. To confirm the biologically significant interaction with SYCE1, the authors should conduct co-immunoprecipitation analyses using isolated germ cells or, at least, testis lysates.
3. Critical mutants are lacking in the analysis of co-dependency for deposition of CE proteins. If available, authors should include analysis of SIX6OS1 in Syce1 and Sycp1 mutants.
4. The authors begin with the premise that in humans the C14orf39/SIX6OS1 gene influences meiotic recombination rate. We also know that recombination rate is dependent on length of SC. Is there any evidence that SIX6OS1 influences length of the SC? For example, are there differences in autosomal SC length in mice homozygous versus heterozygous for the Six6os1 mutation?
5. The counts of RAD51 in early meiotic prophase (p. 14, top paragraph) would provide an interesting view of early onset of recombination. They are reported as "data not shown," but could be incorporated into Supplemental Table 1.
6. The observation that the Six6os1 mutant spermatocytes condense individual chromosomes after OA treatment is surprising. This is because it is generally thought that competence for condensation arises after mid-pachynema, in spermatocytes positive for a sex body, for MLH1 and for histone H1t; yet the mutant spermatocytes do not exhibit MLH1 foci (do they accumulate histone H1t?). This is worthy of more comment.
7. While it is evident from the data that the sex body does not form in Six6os1 mutant spermatocytes, the authors should consider an alternative interpretation: that SIX6OS1 is required, not directly, but indirectly for sex body formation. Perhaps it does not form simply because the germ cells are arrested prior to most of the events that mark the assembly of this specialized chromatin domain.

Reviewer #2 (Remarks to the Author)

In their manuscript entitled "The C14ORF39/SIX6OS1 gene associated with human recombination is a novel constituent of the synaptonemal complex essential for mouse fertility", Gomex-H et al. present an in-depth cytological and functional analysis of a previously uncharacterized ORF that has been (based on a separate study) implicated in influencing human recombination rates. This study demonstrates that the SIX6OS1 ORF encodes a bona fide protein that is expressed in meiotic cells. After examining a SIX6OS1-GFP fusion and seeing that this tagged protein colocalizes with synaptonemal complex (SC) on meiotic chromosomes, the authors used antibodies to characterize the co-localization of SIX6OS1 with SC transverse filament protein SYCP1, as well as SC central element and lateral element proteins. They performed immuno-gold electron microscopy to more specifically localize SIX6OS1; they interpret their immuno-EM data to mean that SIX6OS1 localizes specifically to the central element of the SC but I do not think the provided data is convincing enough on its own to warrant this conclusion. More convincing data supporting a role for SIX6OS1 as a central element component is the interaction that the authors demonstrated between SIX6OS1 and the CE protein SYCE1 (see below), and the finding that in mutants lacking SYCE3 (and thus SYCE1), no SIX6OS1 is present on pachytene chromosomes despite a discrete/detectable amount of SYCP1 (transverse filament protein) assembly on chromosomes in the Syce3- mutant. The authors show not only that SYCE1 and SIX6OS1 interact in a yeast two hybrid system, but that SYCE1 can co-immunoprecipitate with SIX6OS1 in a transient transfection system, and that the N terminus of SIX6OS1 but not the C terminus is required for this interaction. Using the transient transfection system the authors also demonstrate that SIX6OS1 does not co-immunoprecipitate with SYCE2, SYCE3, TEX12 or SYCP1 nor the LE protein SYCP3 nor cohesin component REC8. Finally, the authors used differential tagging and co-immunoprecipitation to show that SIX6OS1 likely exists as a homooligomer. The authors create a mouse line missing SIX6OS1 protein and show that this mutant mouse (whether male or female) is sterile, with arrested spermatogenesis in male mice and depleted follicles in female mice reproductive tissue. In the mutant mice lacking SIX6OS1, meiotic chromosomes do not synapse properly, consistent with the putative role of SIX6OS1 as a structural component of the SC. Meiotic axis-associated proteins such as REC8, STAG3, RAD21L, SMC1B and SYCP3 are present on meiotic chromosomes from mice lacking SIX6OS1, but SYCP1 is nearly completely absent from chromosome axes when SIX6OS1 is missing. (This observation seems at odds with the presence of SYCP1 but absence of SIX6OS1 on chromosomes in Syce3- mutants.) The authors also show that Mlh1 foci are absent from meiotic chromosomes in mice lacking SIX6OS1, consistent with an absence of crossover recombination events, an expectation for mammalian meiocytes lacking SC. Altogether, the authors provide ample cytological, structural and functional evidence that SIX6OS1 encodes a novel component of mammalian SC, and that this protein interfaces with the SYCE1 protein within the SC.

My critiques of the manuscript in its current form derive mainly from certain confusing statements or observations. I believe that most of my critiques (both major and minor) can be resolved by altering the wording of the narrative.

Major Critiques

1. The authors argue that SIX6OS1 does not localize in the same temporal manner as SYCP1 (the SC transverse filament component): "Thus, SIX6OS1 partially overlaps the distribution of SYCP1 at the synapsed axes but not strictly in the same way since SYCP1 appears earlier and disappears later from the synapsed chromosomes", but the data provided do not sufficiently support this claim. From the images of distinct stages of meiosis (Figure 1d), there is no evidence that SYCP1 appears earlier or disappears later than SIX6OS1. If there is data that can support this claim, it needs to be presented. Otherwise, the claim should not be made.
2. The authors claim that SIX6OS1 localizes less robustly than SYCP1 at telomeres and cite Figure 1d (page 6). Perhaps a zoomed in image indicating the basis for this conclusion would help me to be convinced of this claim. Also, how do the authors reconcile this conclusion with the tight co-localization between SYCP1 and SIX6OS1 observed at telomeres in the fluorescence profiles measured along chromosome axes in Figure 2?
3. Text and Figure 1f: "the gold particles exclusively localized to the CE of the SC". The immunogold electron microscopy data provided is not robust or rigorous enough to warrant the

conclusion that SIX6OS1 localizes exclusively to the CE. From the two images alone it is not clear that the localization of these gold particles are CE-specific. In both images, doesn't it look like gold particles are detected in relative abundance at the LE/AEs? More images should be supplied to convince the reader that this protein is CE-specific, and a control experiment (which leaves the primary antibody out) should be examined in order to understand which labeling is more likely to be non-specific binding by the secondary antibody. (Also the size of the gold particles should be noted in the legend.)

4. Figure 4 and Figure 7: The authors have found that SYCP1 accumulates into detectable structures on meiotic chromosomes when SYCE3 is missing, but that in this Syce3- context, SIX6OS1 is not detectable on the chromosomes (Figure 4b, bottom row, and pg. 10: "in mice lacking the central element protein SYCE3,...SIX6OS1 was not detected despite the presence of some discrete SYCP1 foci"). But the authors separately found that when SIX6OS1 is absent, SYCP1 is not detectable on chromosomes (Figure 7). (None of the CE proteins - SYCE1, 2, 3 or TEX12 - are detectable on chromosomes in the absence of SIX6OS1 either.) I am having difficulty reconciling these observations. If the presence of SIX6OS1 is required for the stable accumulation of SYCP1 onto meiotic chromosomes (as the Six6OS1 mutant suggests), why does SYCP1 readily accumulate into structures on chromosomes, despite the absence of SIX6OS1, in Syce3- mutants?

The paragraph on page 13 that should attempt to resolve this interesting but confusing set of observations is itself somewhat confusing: "Similar to SIX6OS1 mutants, in SYCP1-deficient mice SYCE1-3 and TEX12 cannot be loaded to the SC. In contrast, the lack of any of these central element proteins, and especially SYCE3 and SIX6OS1 (this paper) precludes the normal deposition of SYCP1." First, "In contrast" is not an appropriate lead in to the latter sentence, since in both cases the absence of one protein leads to the absence of another... instead of contrasting, the two phenomena are similar.

Second, this statement that any central element protein deficiency precludes the normal deposition of SYCP1 does not address the confusing observation that some SYCP1 does localize in some CE-deficient mutants, even when SIX6OS1 is absent, but almost none is present in SIX6OS1-deficient mutants. An earlier statement and data in Figure 4 indicate that "discrete SYCP1 foci" are detectable in Syce3- mutants, and on page 12, "... double immunolocalized SYCP3 and SYCP1 and show that in contrast to other mutants of the CE as SYCE2 or TEX12, SIX6OS1 deficient spermatocytes have almost undetectable levels of SYCP1 labeling..." (this statement implies that other mutants missing CE proteins exhibit detectable SYCP1 structures on chromosomes).

In the end, it is not clear to the reader whether the authors believe that the mutants lacking SIX6OS1 exhibit a more dramatic deficiency of SYCP1 on chromosomes relative to mutants lacking any of the other CE proteins. This should be clarified in this section of the Results. Furthermore, the discussion here should reconcile the apparently conflicting observation that absence of SIX6OS1 in Syce3- mutants allows SYCP1 structures to accumulate on chromosomes, whereas the absence of SIX6OS1 (and SYCE3) in Six6OS1 mutants results in a block to SYCP1 accumulation on chromosomes.

5. On page 7, the authors characterize Sororin as "a recent cohesin subunit localized to synapsed regions". Supplementary figure 6 shows the localization of SMC3, SMC1beta, and other cohesin-associated proteins, including Sororin, on meiotic chromosomes from mice lacking SIX6OS1. The legend states: "In the absence of SIX6OS1 the levels or distribution of cohesin subunits is not altered". However, the images in Suppl. Fig. 6 indicate that there is a dramatic loss of Sororin in Six6OS1 mutants, relative to wild-type. This observation needs to be addressed in the narrative, since the phenotype seems to be a departure from what was expected.

Minor critiques:

1. (Abstract) "as anonymous gene variant".. better "as an anonymous gene variant"
2. (Abstract) "Sequence analysis reveals an N-terminal alpha helical domain"... more accurate to say "sequence analysis (or modeling) predicts..."
3. (Abstract) "immediately prior to reciprocal recombination and crossover formation" As far as I know, reciprocal recombination and crossover formation are the same thing. Thus it would be most

rigorous to choose one or the other term and not include both, so as not to imply that reciprocal recombination and crossover formation are two distinct phenomena.

4. (Pg. 3) "Recently, by exploiting data resources in Iceland called over two million..." This sentence is missing at least one noun - needs to be reworded to make the narration clear.

5. (Pg. 3) "including the histone acetyl transferase PRDM9" - This is inaccurate; PRDM9 is a histone H3K4 trimethyltransferase.

6. (Pg. 3) "The SC is a proteinaceous structure that connects the homologous chromosomes" I think what is meant is homologous chromosomes.

7. (Pg. 3) "It is known from mouse mutants and ...that alterations in these genes (i.e. STAG3 and SYCE1)..." Since STAG3 was not specifically mentioned in the preceding sentences, it isn't clear how STAG3 fits among the "these genes" you discussed. It would be clearer if the sentence included "(i.e. the meiosis-specific cohesin subunit STAG3 and SYCE1)"

8. (Pg. 4) "To gain further insight into the biological process affecting recombination rates... and given that one of our genes of interest was found as a target in a recent genetic analysis (RAD21L), we investigated the list of novel genes with recently identified coding variants." This sentence needs to be clarified. First, it isn't clear why RAD21L was/is a gene of interest to the authors - is it solely because RAD21L has been linked to an activity that affects recombination rates or was RAD21L of interest to these authors before this link was discovered? The reason why RAD21L is a gene of interest to the authors should be included in the rewritten sentence(s), or not stated at all. Second, "a recent genetic analysis" is overly vague and thus does not help the reader understand the motivation behind the questions that the authors will be addressing. Are they referring to the genetic analysis described in paragraph 2 of the intro? This needs clarified since paragraph 4 is well separated from paragraph 2, interrupted by discussion of the SC structure. Third (and perhaps related to the other issues) "we investigated the list of novel genes with recently identified coding variants" is also too vague. Is there just one list of novel genes with coding variants? (I would think that nearly all genes have coding variants.) What are the more specific criteria that were used to identify genes for this list? Is this list from the same study that linked RAD21L to variation in meiotic recombination (described in paragraph 2)?

9. (Pg. 5) "...the key meiotic histone acetyl transferase PRDM9..." - this is inaccurate - PRDM9 has trimethyltransferase activity, not acetyltransferase activity.

10. (Pg. 5) "...the mismatch repair protein MSH4..." - MSH4 is a member of the MutS mismatch repair gene family - it would be most rigorous to specify this, since MSH4 does not necessarily have mismatch repair activity.

11. Figure 1A and the text: what does "RT" in RT-qPCR stand for (real time, or reverse transcriptase?)

12. Figure 1A: mRNA expression is calculated and the Y axis indicates that "relative expression" is plotted. Are these values the amount of mRNA detected relative to another gene's mRNA (in which case please indicate the gene being used to normalize the expression)? Or are these values reflecting the relative enrichment of SIX6OS1 mRNA in one tissue versus another tissue (in which case the Y axis should simply indicate "Expression (arbitrary units)")

13. Figure 1: The legend indicates that SIX6OS1 localizes with the PAR of the XY bivalent (b and c). In the image please indicate the PAR with an arrowhead or asterix.

14. Pg. 9 "a highly helical structures region" (structural I think?)

15. Pg. 12 "...in contrast to other mutants of the CE as SYCE2 or TEX12 24,37, SIX6OS1 deficient spermatocytes have almost undetectable levels of SYCP1 labeling ..." The reference for TEX12 here is not the appropriate one, since this particular study did not examine SYCP1 label in Tex12-mutants.

16. Pg. 14 "just before reciprocal recombination and CO formation take place" - redundant; choose either reciprocal recombination or crossover formation.

17. Pg. 18 heading typo ("completion")

Reviewer 1

1. There is no overt validation of the antibody recognizing SIX6OS1, although Fig. 5c shows that it recognizes a 70 kD band and is absent in the mutant mice. The Materials and Methods should include a statement on how the antibody was developed and validated, and also a complete gel should be shown (does it recognize only a single band?).

We have included a novel Supplementary Fig. 3 showing the characterization of the antibody we used in the first MS, a commercial Goat polyclonal antibody against C14orf39/SIX6OS1, which was developed by Santa Cruz (sc-245304). This antibody was raised against a conserved peptide mapping within an internal region of human SIX6OS1. We have also included in the Methods section this information: "Goat polyclonal antibodies against C14orf39/SIX6OS1 were developed by Santa Cruz (sc-245304) and used indistinctly for the IF and western blot. This antibody was raised against a conserved internal region of human SIX6OS1 (see Supplementary Fig. 3 for validation)."

To experimentally validate the goat antibody against SIX6OS1 for IF, we transfected HEK 293 cells with an expression plasmid encoding EGFP-SIX6OS1. The transfected cells were analyzed by double IF to detect EGFP (green) and SIX6OS1 using the goat anti-SIX6OS1 (red). The results show co-localization of the red and green labelling in transfected cells.

In addition, the transfected cells were also analyzed by Western Blot using the goat anti-EGFP and the goat anti-SIX6OS1 antibodies. The results show the detection of a band of the expected molecular weight of EGFP + SIX6OS1 (32+70) (Supplementary Fig. 3a).

In order to further validate our results obtained with the goat anti-SIX6OS1 antibody, we made use of a second commercial antibody generated in rabbit with a different antigen (fusion protein of GST with C-350 aa of human SIX6OS1) and affinity purified. The corresponding description has been incorporated in Methods: ". Rabbit polyclonal antibodies against SIX6OS1 were developed by Proteintech™ (22664-1-AP) against a fusion protein of GST with SIX6OS1 (C-350 aa) of human origin (see Supplementary Fig. 3 for validation)"

We then carried out a similar assay in HEK 293 cells with the same expression plasmids (EGFP-SIX6OS1). As can be seen in Supplementary Fig. 3b, the transfected cells were analyzed by double IF to detect EGFP (green) and SIX6OS1 using the rabbit anti-SIX6OS1 (red). The results show again co-localization of the red and green labelling in transfected cells, demonstrating the reactivity of the antibody against mouse SIX6OS1 protein (Supplementary Fig. 3b).

Following the same schedule, we used the transfected cell extracts to check the ability of the new rabbit antibody to detect the overexpressed EGFP-SIX6OS1. The results obtained are not very elegant due to the appearance of several unspecific bands at the same molecular weight where EGFP-SIX6OS1 is detected with the anti-GFP

(supplementary figure 3b). We thus conclude that the new anti-rabbit is not a very useful tool for western blot analysis but demonstrates its reactivity.

Finally and most importantly, we analyzed the expression of the endogenous SIX6OS1 protein in spermatocytes spreads from wild-type and *Six6os1*^{-/-} using the novel rabbit anti-SIX6OS1 antibody using SYCP3 as marker. The results obtained show SIX6OS1 labelling from zygonema to pachynema along the synapsed LEs but **a complete lack of labelling** in the *Six6os1*^{-/-} spermatocytes.

This result is similar to the one obtained with the goat anti-SIX6OS1 in wt (specific labelling, Fig. 1c-d) and KO spermatocytes (complete lack of labelling, Fig. 5d). These results together with the **co-localization** (central element of the SC) of the SIX6OS1-EGFP after *in vivo* electroporation of mouse testis with the endogenous localization of Six6OS1 with the two different antibodies (raised in rabbit and goat with different antigens), demonstrate that these two antibodies recognize the same endogenous SIX6OS1 protein by IF in a very specific manner.

We have been unable to reproduce the western blot analysis of extracts from wt and ko testis using the new rabbit polyclonal antibody against Six6OS1. Accordingly, and given the consistent and robust results obtained also with the new second antibody in Immuno fluorescence analysis of the endogenous SIX6OS1 protein, we have decided to eliminate the western blot analysis of testis extracts and include the new Immuno fluorescence analysis with the new rabbit antibody leading to a new panel in the Supplementary Fig. 3c: “(c) Double immunofluorescence of spermatocytes at pachytene stage obtained from *Six6os1*^{+/+} and *Six6os1*^{-/-} mice using the polyclonal rabbit antibody α -SIX6OS1 (green) and mouse α -SYCP3 (red). The experiments were reproduced three times”. We have also included in the main text the use to two independent antibodies: “In addition, we carried out a detailed analysis of mouse spermatocytes and oocytes spreads through double-labelling with specific antibodies against SIX6OS1 (which were intensively validated, Supplementary Fig. 3 and Fig. 5d)” and also in the section related with the characterization of the KO mouse: “Spermatocytes from homozygous targeted mice did not show SIX6OS1 protein expression by immunofluorescence analysis with two independent polyclonal antibodies (Fig. 5c-d and Supplementary Fig. 3c).

2. The section "Sequence analysis of SIX6OS1" seems out of place and should be included within the first section of Results ("C14orf39/SIX6OS1 is a novel protein composing the CE of the mammalian SC").

We agree and appreciate the suggestion. We have removed the heading “Sequence analysis” and have included this paragraph (reduced version) into the indicated position in the text.

3. Protein interaction studies are comprehensive. However, they are limited to *in vitro* interactions in transiently transfected HEK293 cells. To confirm the biologically significant interaction with SYCE1, the authors should conduct co-immunoprecipitation analyses using isolated germ cells or, at least, testis lysates.

We have tried since the start of this project (functional analysis of SIX6OS1) to immunoprecipitate endogenous SIX6OS1 using different protocols that have worked to IP other proteins from the AEs (Gutierrez-Caballero et al., Cell Cycle, 2011). However, we have been unable to successfully IP the endogenous SIX6OS1 using the two antibodies (goat and rabbit). In an attempt to provide new evidences for a direct interaction between SYCE1 and SIX6OS1 (aside from the reciprocal IP after transfection in HEK 293 cells, co-localization in Cos7 by IF and the unbiased Yeast two hybrid assay in a testis library) we performed Proximity Ligation Assay (PLA) on co-transfected cells with SIX6OS1 and SYCE1. The results confirmed the interaction by IF and IP analysis and indicate a closed proximity (less than 40 nm) between both proteins in the cytoplasm of transfected COS7 cells. The corresponding analysis has been included in the text “We further validated the interaction between SYCE1 and SIX6OS1 in transfected COS7 cells by Proximity Ligation Assay (PLA) (Supplementary Fig. 7)”, and in the new panel of the Supplementary Fig. 7 and in the “Methods” section.

3. Critical mutants are lacking in the analysis of co-dependency for deposition of CE proteins. If available, authors should include analysis of SIX6OS1 in Syce1 and Sycp1 mutants.

We are aware that the use of additional mutants of the central element would be of interest for the analysis of co-dependency. Thus we have tried to get access to these mutant mice. Unfortunately, the SYCE1 mutants are not available and to the best of the knowledge of several labs to which we have contacted (including that of Dr. Ewelina Bolcu-filas at JAX which is the first author of the paper where they generated and analysed the SYCE1 mutant; Bolcun-Filas et al., PLoS Genetics 2009) these strain has been lost because of the retirement of Dr. Cooke (senior author of the paper). In any case, we believe that we can predict with the analysis of SYCE3 mutants [in which SYCE1 is not loaded according to Schram et al., 2009 PLOS Genetics; and the reviewer2: ” and the finding that in mutants lacking SYCE3 (and thus SYCE1), no SIX6OS1 is present on pachytene chromosomes”] that SIX6OS1 would not be present on pachytene chromosomes. We have included this argumentation in the text: “In this same regard, we predict that SYCE1 mutants will also be defective in SIX6OS1 loading since mutants lacking SYCE3 (and thus SYCE1³²) no SIX6OS1 is detected onto the LEs (Fig. 4)”.

However, we have analyzed the SYCP1-deficient spermatocytes that the reviewer indicated. The results show complete loss of loading of SIX6OS1 in the SYCP1 null mice. The results have been incorporated in a new panel of Fig. 4 and in the main text: “Finally, in mice lacking the central element proteins SYCE3 and SYCP1, in which AEs completely fail to synapse in a pachytene-like stage^{32,33}, SIX6OS1 was not detected despite the presence of a weak discontinuous pattern of SYCP1 deposition in the SYCE3 mutant (Fig. 4 and Supplementary Fig. 9)”

4. The authors begin with the premise that in humans the C14orf39/SIX6OS1 gene

influences meiotic recombination rate. We also know that recombination rate is dependent on length of SC. Is there any evidence that SIX6OS1 influences length of the SC? For example, are there differences in autosomal SC length in mice homozygous versus heterozygous for the *Six6os1* mutation?

This is an interesting point but difficult to undertake experimentally because the length of the axis can be affected by the lack of synapsis. Thus and although we can measure them there is no a warrantee. Anyway, we have performed the analysis and the results show that there are no statistical differences between *Six6os1* +/+ , *Six6os1* +/- and *Six6os1* -/- by measuring and comparing the total length of the autosomal axes. We have not included this info in the revised MS to avoid an increase in the length of the MS and because of the negative result.

5. The counts of RAD51 in early meiotic prophase (p. 14, top paragraph) would provide an interesting view of early onset of recombination. They are reported as "data not shown," but could be incorporated into Supplemental Table 1.

We have included the counts of RAD51 in the new supplemental table 2, as suggested by the reviewer and also a short sentence has been included in the results. "During early stages (leptonema), RAD51 distribution in mutants were similar to wild type controls (Supplementary Table 2). However, in *Six6os1*^{-/-} spermatocytes at zygotene and pachytene-like stage, both RAD51 and RPA remained partially associated with the AEs (Fig. 8a; see Supplementary Table 1 and 2)"

6. The observation that the *Six6os1* mutant spermatocytes condense individual chromosomes after OA treatment is surprising. This is because it is generally thought that competence for condensation arises after mid-pachynema, in spermatocytes positive for a sex body, for MLH1 and for histone H1t; yet the mutant spermatocytes do not exhibit MLH1 foci (do they accumulate histone H1t?). This is worthy of more comment.

We agree that H1t staining can delineate more precisely the sub-stage of the very long pachytene stage at which the spermatocytes are arrested. Following his/her comment, we have analyzed if H1t has been incorporated to the arrested spermatocytes in mice lacking SIX6OS1 and show that they are positive for H1t staining. We have included this novel data in the text and in the Supplementary Fig. 10c: “Finally, and to refine the stage of the blockade, we immunolabelled Six6os1^{-/-} spermatocytes with the mid pachytene-specific histone variant H1t. The positive staining for H1t (Supplementary Fig. 10c) indicates that arrested spermatocytes reach the mid-pachytene stage.” In addition we have also incorporated a similar explanation in the OA-section “To further validate this, and in light of the late arrest at mid-pachytene-like stage (H1t positive)”.

7. While it is evident from the data that the sex body does not form in Six6os1 mutant spermatocytes, the authors should consider an alternative interpretation: that SIX6OS1 is required, not directly, but indirectly for sex body formation. Perhaps it does not form simply because the germ cells are arrested prior to most of the events that mark the assembly of this specialized chromatin domain.

In the original MS we stated that “These results indicate that SIX6OS1 is essential for the formation of the sex body”. We did not discuss or mention if this was a direct or indirect effect. We agree with the reviewer that it could be required indirectly as most of the asynaptic mutants are unable to assemble a sex body. However, and given the positive H1t staining in the arrested spermatocytes at mid-pachytene when sex body is already assembled, we believe that “timing” would not be a satisfactory explanation since the germ cells are not arrested prior to sex body assembly. We have thus modified the general conclusion in the MS as follows “Together, these results indicate that SIX6OS1 deficiency, similar to most asynaptic mice mutants, indirectly blocks sex body formation”.

Reviewer 2

1. The authors argue that SIX6OS1 does not localize in the same temporal manner as SYCP1 (the SC transverse filament component): “Thus, SIX6OS1 partially overlaps the distribution of SYCP1 at the synapsed axes but not strictly in the same way since SYCP1 appears earlier and disappears later from the synapsed chromosomes”, but the data provided do not sufficiently support this claim. From the images of distinct stages of meiosis (Figure 1d), there is no evidence that SYCP1 appears earlier or disappears later than SIX6OS1. If there is data that can support this claim, it needs to be presented. Otherwise, the claim should not be made.

We appreciate the comment and realized that in the original state and without the required explanations, the claim is not justified. Bearing on mind that we have already

exceeded the limit of words by far and that the observation does not add essential information we have decided to remove such claim from the text.

2. The authors claim that SIX6OS1 localizes less robustly than SYCP1 at telomeres and cite Figure 1d (page 6). Perhaps a zoomed in image indicating the basis for this conclusion would help me to be convinced of this claim. Also, how do the authors reconcile this conclusion with the tight co-localization between SYCP1 and SIX6OS1 observed at telomeres in the fluorescence profiles measured along chromosome axes in Figure 2?

We have analyzed the data and figures and we agree with him/her that there is no any zoomed in image showing that SIX6OS1 is diminished at the telomeres. Accordingly, we have included a new zoom in image. However, as the reviewer points out this fact (less SIX6OS1 at the telomeres) is not reflected at the fluorescence profiles as it should be.

Thus, we have also re-examined the fluorescence profiles to find an explanation. The reason is that we used the SIX6OS1 profile as the template onto which we compared the SYCP1 fluorescence profile. Thus, the telomeric subregions were under-evaluated or not considered at all. We have now inverted the analysis and consequently the "telomeric distortion" is eliminated and, accordingly, in the new profile telomeres are now regions where SIX6OS1 is less abundant. Consequently, we have included in the text a new call to the new figure 2 to show this effect and the corresponding zoom in figure 1d.

3. Text and Figure 1f: "the gold particles exclusively localized to the CE of the SC". The immunogold electron microscopy data provided is not robust or rigorous enough to warrant the conclusion that SIX6OS1 localizes exclusively to the CE . From the two images alone it is not clear that the localization of these gold particles are CE-specific. In both images, doesn't it look like gold particles are detected in relative abundance at the LE/AEs? More images should be supplied to convince the reader that this protein is CE-specific, and a control experiment (which leaves the primary antibody out) should be examined in order to understand which labeling is more likely to be non-specific binding by the secondary antibody. (Also the size of the gold particles should be noted in the legend.)

In relation with this point, and to attend the reviewer's concerns, we agree with him/her and would like to say that we always incorporate a negative control without primary antibody in the immunostaining and especially in the immune-gold electron microscopy although such a control is usually not shown in the MSs or papers. We agree that the term "exclusively" is not strict enough given the dispersion of the signal when using immune-gold electron microscopy. Consequently, we have modified in the main text of the MS this term and substituted by a more realistic description "the gold particle distribution supports localization at the CE of the SC". We would like to show to the reviewer the figure corresponding to the negative control.

4. Figure 4 and Figure 7: The authors have found that SYCP1 accumulates into detectable structures on meiotic chromosomes when SYCE3 is missing, but that in this *Syce3*⁻ context, SIX6OS1 is not detectable on the chromosomes (Figure 4b, bottom row, and pg. 10: "in mice lacking the central element protein SYCE3,...SIX6OS1 was not detected despite the presence of some discrete SYCP1 foci"). But the authors separately found that when SIX6OS1 is absent, SYCP1 is not detectable on chromosomes (Figure 7). (None of the CE proteins - SYCE1, 2, 3 or TEX12 - are detectable on chromosomes in the absence of SIX6OS1 either.) I am having difficulty reconciling these observations. If the presence of SIX6OS1 is required for the stable accumulation of SYCP1 onto meiotic chromosomes (as the *Six6os1* mutant suggests), why does SYCP1 readily accumulate into structures on chromosomes, despite the absence of SIX6OS1, in *Syce3*⁻ mutants?

We appreciate the point, and we think that part of the controversy could be due to the use of misleading terms (almost undetectable) instead of the exact value that already was present in the Supplementary Fig. 9c (quantitation of the SYCP1 signal). We have thus included in the text the numerical comparison and modify the description to adjust it more realistically to the numerical values.

From the biological point of view, there is no an easy explanation for this differences in the SYCP1 labelling between *Six6os1*^{-/-} (93,7% reduction of staining, that is 6.3% of positive staining) and *Syce3*^{-/-} (54,36% reduction, that is 45.7% of positive staining). Anyway, these values are far from being presence/absence.

From our genetic data, it seems that the SC is more complex than expected and that interactions between subunits are not linear but occur in several dimensions. Until recently, the expected interaction between SYCP1 and SYCE3 according to the genetic mutants, was unvalidated biochemically and instead it was thought that SYCP1 was interacting with SYCE1. We now know from biochemical methods that SYCE3 is the true partner of SYCP1. Thus, from our genetic analysis of SIX6OS1 deficiency it cannot rule out that SIX6OS1 could also be interacting with SYCP1 in addition to SYCE1. This interaction would explain the lower staining of SYCP1 in the SIX6OS1 KOs than in the SYCE3 KOs. However, this is of course extremely speculative and would need further biochemical support and is at present one of the research line of the Owen's lab in a collaborative manner.

The paragraph on page 13 that should attempt to resolve this interesting but confusing set of observations is itself somewhat confusing: "Similar to SIX6OS1 mutants, in SYCP1-deficient mice SYCE1-3 and TEX12 cannot be loaded to the SC. In contrast, the lack of any of these central element proteins, and especially SYCE3 and SIX6OS1 (this paper) precludes the normal deposition of SYCP1." First, "In contrast" is not an appropriate lead in to the latter sentence, since in both cases the absence of one protein leads to the absence of another... instead of contrasting, the two phenomena are similar.

We agree and appreciate the comment, the sentence is not well constructed since as the reviewer points there is no contrasting and the two phenomena are similar. We used this paragraph to try to explain the paradoxical observation and unfortunately we did not use the appropriate term. We appreciate the comment and we have accordingly substituted the term "in contrast" by "surprisingly", since in the reciprocal mutants the absence of the protein also lead to the absence of the others. Thus, we have reorganized the whole sentence trying to properly communicate that this paradox is not restricted to SIX6OS1 but is a general observation within the CE family of proteins.

Second, this statement that any central element protein deficiency precludes the normal deposition of SYCP1 does not address the confusing observation that some SYCP1 does localize in some CE-deficient mutants, even when SIX6OS1 is absent, but almost none is present in SIX6OS1-deficient mutants. An earlier statement and data in Figure 4 indicate that "discrete SYCP1 foci" are detectable in Syce3- mutants, and on page 12, "... double immunolocalized SYCP3 and SYCP1 and show that in contrast to other mutants of the CE as SYCE2 or TEX12, SIX6OS1 deficient spermatocytes have almost undetectable levels of SYCP1 labeling..." (this statement implies that other mutants missing CE proteins exhibit detectable SYCP1 structures on chromosomes). In the end, it is not clear to the reader whether the authors believe that the mutants lacking SIX6OS1 exhibit a more dramatic deficiency of SYCP1 on chromosomes relative to mutants lacking any of the other CE proteins. This should be clarified in this section of the Results. Furthermore, the discussion here should reconcile the apparently conflicting observation that absence of SIX6OS1 in Syce3- mutants allows SYCP1 structures to accumulate on chromosomes, whereas the absence of SIX6OS1 (and SYCE3) in Six6OS1 mutants results in a block to SYCP1 accumulation on chromosomes.

We understand the point raised by the reviewer but we would like to mention that we have tried to avoid the term block or absence of SYCP1 in the SIX6OS1 mutants, instead we use the quantification values of the SYCP1 signal that was already present in the original MS (Supplementary Fig. 9c). However, as we can deduce from the concerns raised by the reviewer, our mistake was to convert not accurately the numerical differences in words within the text. We agree that the term employed "almost undetectable" is not a very suited quantitative expression and is too vague to

communicate that SYCP1 deposition is more reduced in SIX6OS1 (93% reduction) than in SYCE3 (54% reduction). Accordingly, we have included this numerical data from the Supplementary Fig. 9c to the main text in order to clarify the observation: “In contrast to other CE mutants such as SYCE3, and specially SYCE236 and TEX1239, SIX6OS1 deficient spermatocytes have reduced levels of SYCP1 labelling (93,70% reduction in Six6os1-/- vs 54,36 % reduction in Syce3-/-). Mutant oocytes, however, show a slightly weaker reduction of SYCP1 staining (79,28% reduction, Fig. 7a-b and Supplementary Fig. 9c for quantitation”. This difference cannot be mechanistically explained and is accordingly mentioned in the discussion. We believe that this point is not a conflict since there is no any block of SYCP1 accumulation in the absence of SYCE3 but just a strong reduction.

We have avoided the use of “normal deposition” and defined it in positive terms. We also state that the SIX6OS1 has a stronger phenotype: “In SYCP1-deficient mice, SIX6OS1 (Fig. 4), SYCE1-3 and TEX12 cannot be loaded to the SC. Surprisingly, the lack of any of these central element proteins leads to the aberrant deposition of SYCP1 in a weak discontinuous pattern, with the severest phenotype occurring in SIX6OS1 deficiency (weakest staining)” to clearly distinguish it from the continuous pattern connecting synapsed chromosomes in wt spermatocytes.

From a speculative point of view, it cannot ruled out, however, that very low amounts of SIX6OS1 could be loaded in the absence of SYCE3 (undetectable by eye) and that they could be responsible of the differences in the SYCP1 labelling. Alternatively, we are considering that the SYCP1 labelling in both SIX6OS1 and SYCE3 mutants are of the same nature. However, they could underlie different processes such as the RPA accumulation before RAD51 loading in some meiotic mutants (double kleisin mutants. Llano et al., JCB, 2012) which is different from the RPA accumulation that occurs in most meiotic mutants with a pachytene arrest in which the RPA loading occurs after RAD51 loading.

5. On page 7, the authors characterize Sororin as "a recent cohesin subunit localized to synapsed regions". Supplementary figure 6 shows the localization of SMC3, SMC1beta, and other cohesin-associated proteins, including Sororin, on meiotic chromosomes from mice lacking SIX6OS1. The legend states: "In the absence of SIX6OS1 the levels or distribution of cohesin subunits is not altered". However, the images in Suppl. Fig. 6 indicate that there is a dramatic loss of Sororin in Six6OS1 mutants, relative to wild-type. This observation needs to be addressed in the narrative, since the phenotype seems to be a departure from what was expected.

We do appreciate the criticism raised by the reviewer and agree with him/her that the Sororin distribution is clearly affected in the *Six6os1* mutant, as it is largely expected from protein strictly localized to the synapsed LE (although with an unknown function; Gomez et al., EMBO Report 2016). Given that the *Six6os1* mutant is deficient in synapsis, the loss of SIX6OS1 leads to the loss of Sororin loading as we have shown in the MS for other CE proteins such as SYCE1-3, and TEX12. The reason for this error is likely due to the “expected” observation and thus we unfortunately did not mention it

properly in the legend to Supplementary Fig. 8. We have corrected the error in the text: “Similarly, the regulatory cohesin subunit Sororin, which is located at the CE, is also lacking in the SIX6OS1 deficient spermatocytes (Supplementary Fig. 8c).” and legend: “whereas Sororin co-localizes to synapsed LEs and the pseudoautosomal synapsed region of the XY bivalent. In the absence of synapsis in Six6os1-/- spermatocytes, the levels of distribution of cohesin subunits SMC1 β , SMC3, STAG3, REC8 and RAD21L are not altered, whereas Sororin is not loaded, as expected for a cohesin located at the CE of the SC.”

Regarding the minor critiques raised by the reviewer 2, we do appreciate the detailed correction of the text and we agree with all the suggestions.

Reviewer #1 (Remarks to the Author)

General Comments:

The revised manuscript addresses the concerns that I raised in the previous review.

Unfortunately, the manuscript will require serious editing for English language usage.

Additionally, human and mouse gene and protein designation and text formatting is not correct, even in the Abstract, where statements are misleading because of incorrect usage/text formatting. Please bear in mind that it is *SIX6OS1* (or *C14ORF39*) italicized, for the human gene; *Six6os1*, italicized, for the mouse gene; and *SIX6OS1*, not italicized, for the protein in either species. (For further detail and explanation, please see appropriate nomenclature sections of HUGO and MGI databases.)

Specific Comments:

Line 305: this refers to Fig. 5c-d, but there is no panel d in Fig. 5

Line 461: perhaps "impedes sex body formation" would be better than "indirectly blocks sex body formation"

Line 566-567: What does "indistinctly" mean in this sentence (see just below)? Does it mean that the antibody was used for both IF and western analyses?

"Goat polyclonal antibodies against C14orf39/SIX6OS1 were developed by Santa Cruz (sc-245304) and used indistinctly for the IF and western blot."

Fig. 4 Legend: should be "... AE proteins" (not "... AEs proteins")

Reviewer #2 (Remarks to the Author)

In their revised manuscript entitled "The C14ORF39/SIX6OS1 gene associated with human recombination is a novel constituent of the synaptonemal complex essential for mouse fertility", Gomez-H et al. present an in-depth cytological and functional analysis of a previously uncharacterized ORF that has been (based on a separate study) implicated in influencing human recombination rates. The authors provide ample cytological, structural and functional evidence that *SIX6OS1* encodes a novel component of mammalian SC, and that this protein interfaces with the SYCE1 protein within the SC.

In my opinion, the critiques given for the first version manuscript (by myself and by the other reviewer) have been sufficiently addressed in the new version.

One minor critique: Now that the manuscript has been rearranged somewhat, line 174 should be modified; there doesn't seem to be enough data, so early in the manuscript narrative and before any sub-cellular distribution of *SIX6OS1* is shown, to warrant an outright prediction that this uncharacterized protein interfaces with structural proteins of the SC. The idea could be toned down to a "speculation", coming at the end of this paragraph.

Point-by-point response

Reviewer 1

Unfortunately, the manuscript will require serious editing for English language usage.

→ We have used two independent (English native) scientists from the field (which are often reviewers in related journals) to check the English style of the revised MS. We are now sure that the MS fits to the most rigorous grammar standards.

Additionally, human and mouse gene and protein designation and text formatting is not correct, even in the Abstract, where statements are misleading because of incorrect usage/text formatting. Please bear in mind that it is SIX6OS1 (or C14ORF39) italicized, for the human gene; Six6os1, italicized, for the mouse gene; and SIX6OS1, not italicized, for the protein in either species. (For further detail and explanation, please see appropriate nomenclature sections of HUGO and MGI databases.)

→ We have revised again the whole MS by two blinded co-authors and those discrepancies have been discussed to gain an agreement. Accordingly, the changes have been made in the MS. Please, note that there are some situations in which both options are plausible (gene or protein). In those situations, please use either term that the editorial policy might consider most suited.

Specific Comments:

Line 305: this refers to Fig. 5c-d, but there is no panel d in Fig. 5.

→ The reference to Fig. 5d has been eliminated in the MS.

Line 461: perhaps "impedes sex body formation" would be better than "indirectly blocks sex body formation".

→ The change has been introduced in the new MS.

Line 566-567: What does "indistinctly" mean in this sentence (see just below)? Does it mean that the antibody was used for both IF and western analyses?

"Goat polyclonal antibodies against C14orf39/SIX6OS1 were developed by Santa Cruz (sc-245304) and used indistinctly for the IF and western blot."

→ We have modified this sentence. During the first revision the WBs were already eliminated from the main figures. Thus, we have re-write it in the methods section: "Goat polyclonal antibodies against C14ORF39/SIX6OS1 were developed by Santa Cruz (sc-245304) and used for the immunofluorescence analysis. This antibody was raised against a conserved internal region of human SIX6OS1. Rabbit polyclonal antibodies against SIX6OS1 were developed by Proteintech™ (22664-1-AP) against a fusion protein of GST with SIX6OS1 (C-350 aa) of human origin (see Supplementary Fig. 3 for validation) and was used to validate the immunofluorescence results obtained with the goat polyclonal antibody against C14ORF39/SIX6OS1 developed by Santa Cruz."

Fig. 4 Legend: should be "... AE proteins" (not "... AEs proteins").

→ The change has been introduced in the MS.

Reviewer 2

One minor critique: Now that the manuscript has been rearranged somewhat, line 174 should be modified; there doesn't seem to be enough data, so early in the manuscript narrative and before any sub-cellular distribution of SIX6OS1 is shown, to warrant an outright prediction that this uncharacterized protein interfaces with structural proteins of the SC. The idea could be toned down to a "speculation", coming at the end of this paragraph.

→ We agree with the reviewer and consequently we have modified the text as suggested: "We therefore predict that this N-terminal helical region could mediate interactions with structural proteins of the SC".